# Neurotransmitter Levels (Dopamine, Epinephrine, Norepinephrine, Serotonin) and Associations with Lipid Profiles in Patients with Prediabetes or Newly Diagnosed Type 2 Diabetes Mellitus

**DOI:** 10.3390/ijms262010068

**Published:** 2025-10-16

**Authors:** Roxana Viorela Ahrițculesei, Lidia Boldeanu, Mohamed-Zakaria Assani, Adina Mitrea, Cosmin Vasile Obleaga, Ionela Mihaela Vladu, Diana Clenciu, Mihail Virgil Boldeanu, Cristin Constantin Vere

**Affiliations:** 1Doctoral School, University of Medicine and Pharmacy of Craiova, 200349 Craiova, Romania; roxana.blendea@gmail.com (R.V.A.); mohamed.assani@umfcv.ro (M.-Z.A.); 2Department of Microbiology, Faculty of Medicine, University of Medicine and Pharmacy of Craiova, 200349 Craiova, Romania; lidia.boldeanu@umfcv.ro; 3Department of Immunology, Faculty of Medicine, University of Medicine and Pharmacy of Craiova, 200349 Craiova, Romania; 4Department of Diabetes, Nutrition and Metabolic Diseases, Faculty of Medicine, University of Medicine and Pharmacy of Craiova, 200349 Craiova, Romania; adina.mitrea@umfcv.ro (A.M.); ionela.vladu@umfcv.ro (I.M.V.); dianaclenciu@yahoo.com (D.C.); 5Department of Surgery, University of Medicine and Pharmacy of Craiova, 200349 Craiova, Romania; 6Department of Gastroenterology, University of Medicine and Pharmacy of Craiova, 200349 Craiova, Romania; vere_cristin@yahoo.com

**Keywords:** neurotransmitters, type 2 diabetes mellitus, insulin resistance indices, biomarkers

## Abstract

Neurotransmitters play a pivotal role not only in central nervous system signaling but also in the regulation of systemic energy metabolism, insulin sensitivity, and cardiovascular function. The contribution of neuroendocrine dysregulation to the development of type 2 diabetes mellitus (T2DM) is increasingly being recognized; however, the interplay between neurotransmitter levels and lipid/insulin resistance profiles in T2DM and prediabetes (PreDM) remains poorly characterized. We evaluated serum dopamine (DA), norepinephrine (NE), epinephrine (EPI), and serotonin (ST) in 110 individuals with PreDM (*n* = 40) or newly diagnosed T2DM (*n* = 70). Extended metabolic profiling included HbA1c, lipid panels, and insulin resistance indices (triglyceride-to-glucose index (TyG), TyG-derived indices). Neurotransmitter levels were compared across body mass index (BMI) categories, gender, and glycosylated hemoglobin A1c (HbA1c) quartiles. We applied multivariable linear regression (MLR) adjusted for body mass index (BMI), age, sex, lipids, penalized logistic regression (predicting T2DM status), and exploratory Spearman correlations with False Discovery Rate (FDR) correction. All four neurotransmitters were significantly higher in T2DM versus PreDM (*p* < 0.001). In T2DM patients, DA and NE levels increased across HbA1c quartiles, and NE levels were significantly higher in quartile 3 compared to quartile 2 (*p* = 0.045). In multivariable models, T2DM status was the only consistent predictor of neurotransmitter elevations. Logistic regression identified ST (OR = 8.70) and NE (OR = 3.76) as key discriminators of T2DM status, in addition to HbA1c. Exploratory correlation analyses in T2DM showed trends between EPI and insulin resistance indices (TyG adjusted for waist circumference (TyG-WC), TyG adjusted for waist-to-height ratio (TyG-WHtR)) and between DA and low-density lipoprotein cholesterol (LDL-C), although these did not survive to FDR correction. Neurotransmitter levels are elevated in T2DM and correlate with glycemic and metabolic profiles, suggesting early neuroendocrine involvement in the pathogenesis of diabetes. Serotonin and norepinephrine may serve as adjunctive biomarkers for disease stratification, meriting further prospective and mechanistic investigation.

## 1. Introduction

Type 2 diabetes mellitus (T2DM) and its precursor, prediabetes (PreDM), are significant global health concerns due to their high and increasing prevalence, progressive nature, and strong links to cardiovascular and other metabolic complications [1,2]. T2DM is characterized by chronic high blood sugar levels resulting from a combination of insulin resistance and deteriorating pancreatic β-cell function. Meanwhile, PreDM represents an intermediate phase where fasting plasma glucose, postprandial glucose, or glycated hemoglobin levels are above normal but still below the diagnostic criteria for diabetes [2]. This transitional phase is especially important clinically because it offers a vital window for intervention. According to the International Diabetes Federation (IDF), over 500 million adults currently live with diabetes, and a large portion are in the prediabetic stage, with many progressing to full-blown disease within a decade without targeted preventive actions [1]. The early stages of dysglycemia involve subtle yet biologically meaningful changes in metabolic regulation, inflammation, oxidative stress, and neuroendocrine activity, all of which can influence the disease course and treatment response [3,4].

Neurotransmitters, including dopamine (DA), epinephrine (EPI), norepinephrine (NE), and serotonin (ST), have traditionally been studied in the central nervous system for their roles in regulating reward processing, mood, motivation, stress responses, cognition, and autonomic functions. However, emerging evidence highlights their systemic effects on energy balance, influencing glucose homeostasis, insulin secretion, and lipid metabolism. A recent review highlights that dysregulated neurotransmission may contribute to obesity and T2DM through mechanisms that connect central appetite control with peripheral metabolic signaling [5]. DA influences insulin secretion from pancreatic β-cells and shapes reward-driven eating behaviors [6,7,8]; EPI and NE, secreted primarily from the adrenal medulla and sympathetic nerve endings, promote acute glycemic control through adrenergic receptor-mediated stimulation of glycogenolysis, gluconeogenesis, and lipolysis [9,10]; and ST, synthesized both centrally and in peripheral tissues such as the gut, contribute to satiety signaling, regulation of insulin sensitivity, lipid oxidation, and thermogenesis [11,12,13,14]. Furthermore, serotonergic dysfunction has been implicated in diabetic macroangiopathy and neuropathy, underscoring the clinical relevance of peripheral serotonin alterations [15].

Although previous studies have often focused on individual neurotransmitters or limited metabolic parameters, no research has yet integrated a comprehensive neurochemical profile (DA, NE, EPI, ST) with metabolic, lipid, and insulin resistance indices (triglyceride to glucose index (TyG), TyG adjusted for body mass index (TyG-BMI), TyG adjusted for waist circumference (TyG-WC), TyG adjusted for waist-to-height ratio (TyG-WHtR), and triglycerides to high-density lipoprotein cholesterol (TG/HDL-C)) in populations ranging from PreDM to T2DM. Additionally, cross-sectional studies often fail to account for confounding factors, such as BMI, age, sex, and glycemic control, within multivariable analyses.

In light of these gaps, our study hypothesizes that:

Neurotransmitter serum levels differ significantly between PreDM and newly diagnosed T2DM.Neurochemical changes are linked to lipid and insulin resistance markers beyond traditional glycemic measures.Neurotransmitter profiles could have diagnostic or stratification value in early T2DM classification.

To test these hypotheses, participants were recruited from two university clinical hospitals serving Dolj County, Romania, providing a regionally representative cohort. We measured serum levels of DA, NE, EPI, and ST in subjects with PreDM and newly diagnosed T2DM, and assessed their associations with metabolic, lipid, and combined indices using multivariable regression, penalized logistic models, and correlation analyses. This integrated approach aims to clarify neurochemical signatures of early diabetes and explore their translational relevance for biomarker development.

## 2. Results

### 2.1. Characteristics of Patients with PreDM and T2DM: Clinical and Demographic Overview at Baseline

Based on the data presented in Table 1, several differences and similarities are observed between patients with PreDM and those with T2DM. T2DM patients are generally older (63.87 ± 11.02 vs. 56.18 ± 12.09, *p* = 0.002), reflecting the natural progression of metabolic syndrome toward diabetes with age. However, the gender distribution did not show a significant difference. Regarding the residential distribution of patients, most individuals in both the T2DM and PreDM groups reside in urban areas. Additionally, statistical analysis revealed no significant difference in residential status between the two groups, with a *p*-value of 0.786.

The BMI, waist-to-hip ratio (WHR), and WHtR are similar across groups with no significant difference (*p* = 0.384, *p* = 0.791, and *p* = 0.188, respectively).

Blood glucose levels (fasting and postprandial) and glycosylated hemoglobin A1c (HbA1c) are significantly higher in T2DM patients, confirming the differentiation of glycemic diagnosis: fasting plasma glucose (FPG) (143.00 [106.0–273.0] vs. 103.50 [56.0–121.0] mg/dL, *p* < 0.0001), two-hour plasma glucose after a 75g oral glucose tolerance test (2hPG) (248.00 [129.0–475.0] vs. 151.50 [141.0–196.0] mg/dL, *p* < 0.0001), and HbA1c (9.04 ± 2.16 vs. 5.97 ± 0.22%, *p* < 0.0001).

Regarding renal function, the estimated glomerular filtration rate, as determined by the Modification of Diet in Renal Disease (MDRD) study, indicates significant renal impairment between the groups T2DM vs. PreDM (93.52 ± 30.43 vs. 70.96 ± 26.08%, *p* < 0.0001). Creatinine levels are also comparable between PreDM and T2DM (0.87 [0.56–1.75] vs. 0.74 [0.42–2.35], *p* = 0.0557).

Lipid profile: T2DM was associated with lower HDL-C levels and a higher TG/HDL-C ratio. Differences in total cholesterol (TC) and low-density lipoprotein cholesterol (LDL-C) are minimal and not statistically significant.

Associated lipid indices (TyG, TyG-BMI, TyG-WC, and TyG-WHtR) showed that TyG had a highly significant *p*-value (<0.0001), followed by TyG-WC index (*p* = 0.038). TyG-BMI and TyG-WHtR were similar between the groups, indicating a state of insulin resistance present in both PreDM and T2DM.

Hematology: The white blood cell (WBC) count is higher in patients with T2DM, but these differences did not reach the significance limit (*p* = 0.0577); WBCs indicate ongoing low-grade inflammation, a common feature of T2DM.

Overall, the data indicate that significant metabolic changes occur during the transition from PreDM to T2DM, particularly in terms of glycemic control and lipid levels.

### 2.2. Comparing the Neurotransmitter Values Between the PreDM and T2DM Groups

The biogenic amines measured in our study showed significantly higher levels in individuals with T2DM compared to those in the PreDM group (Table 2). This pattern supports the idea that the neuroendocrine system and the sympathoadrenal axis are involved in the development of diabetes.

The DA value rose from 461.70 pg/mL in PreDM to 865.20 pg/mL in T2DM, a 1.87-fold increase. This notable rise may indicate a compensatory dopaminergic response aimed at sustaining energy balance during hyperglycemia and insulin resistance. ST showed a significant increase from 285.70 pg/mL in PreDM to 987.30 pg/mL in T2DM, a 3.46-fold rise; higher levels could worsen insulin resistance and promote chronic inflammation. Serum EPI levels doubled from 476.90 pg/mL in PreDM to 991.00 pg/mL in T2DM, a 2.07-fold increase, indicating activation of the sympathetic nervous system amidst metabolic disturbances. NE increased by more than 3.71 times, from 2.93 pg/mL to 10.88 pg/mL, representing the most substantial change in this study.

#### 2.2.1. Comparing the Neurotransmitter Levels According to BMI Category in PreDM and T2DM Groups

Table 3 highlights changes in neurotransmitters—DA, ST, NE, and EPI—across BMI categories in both PreDM and T2DM populations.

In PreDM, ST shows a positive trend with BMI, approaching statistical significance (*p* = 0.078). This suggests a potential link between adiposity and serotonergic activity in early dysglycemia.

In T2DM, none of the neurotransmitters differ significantly across BMI groups, possibly due to ceiling effects, metabolic compensation, or more heterogeneity in advanced disease.

DA shows a mild decrease with BMI in PreDM, but not in T2DM. EPI and NE do not appear to be BMI-sensitive in either group.

#### 2.2.2. Comparing Neurotransmitter Levels According to Gender in the PreDM and T2DM Groups

Across both groups, no statistically significant differences in neurotransmitter levels were found between sexes (Table 4).

PreDM: No significant sex-related differences were found for any neurotransmitter. However, females showed higher DA, and males had higher ST, suggesting possible biological trends, but the difference was not statistically significant.

T2DM: Only EPI showed a near-significant (*p* = 0.072) sex difference, with higher levels in females. This may reflect gender-specific autonomic dysregulation in diabetes.

These findings may indicate early-stage sex differences in neuroendocrine responses related to glucose metabolism, particularly under diabetic conditions.

#### 2.2.3. Comparing Neurotransmitter Levels According to the HbA1c Quartiles in the PreDM and T2DM Groups

##### Comparison by HbA1c Quartiles

In Table 5 and Figure 1, the distribution of DA, NE, EPI, and ST across quartiles of HbA1c in patients with PreDM and T2DM was compared. All values are presented as median (range), indicating the use of nonparametric testing (one-way ANOVA (Kruskal–Wallis) to compare groups.

*PreDM patients*—No neurotransmitter shows statistically significant differences across HbA1c quartiles. However, dopamine and epinephrine exhibit noticeable variation, which may become more significant in larger samples.

DA—median values display a U-shaped pattern: highest in Q1 (556.67 pg/mL) and Q4 (557.84 pg/mL), lowest in Q2 (294.40 pg/mL); differences are not statistically significant (*p* = 0.240). EPI—slight decreasing trend from Q1 (550.86 pg/mL) to Q4 (435.88 pg/mL), but not consistent; *p* = 0.124, not significant, though it may indicate subtle variation.

*T2DM patients*—DA and NE vary significantly across HbA1c quartiles. These findings suggest that the progression of diabetes severity is linked to increased catecholaminergic activity and may reflect changes in the autonomic nervous system and dopaminergic tone as glycemic control worsens. EPI and ST appear unrelated to HbA1c levels.

DA shows a gradual increase from Q1 (829.41 pg/mL) to Q4 (1124.30 pg/mL), with a statistically significant difference (*p* = 0.047). This suggests that higher DA levels are linked to worsening glycemic control.

NE increased from Q2 (8.67 pg/mL) to Q3 (14.58 pg/mL), then slightly decreased in Q4 (11.27 pg/mL), with this change being statistically significant (*p* = 0.032). This suggests possible sympathetic activation within moderate HbA1c ranges.

##### Post Hoc Analysis

Given the significant results from the Kruskal–Wallis test in the T2DM group, we applied post hoc Tukey HSD (Honestly Significant Difference) tests to identify specific quartile-level differences. Post hoc Tukey HSD performs pairwise comparisons between all possible combinations of quartiles (e.g., Q1 vs. Q2, Q1 vs. Q3, etc.) and adjusts the *p*-values for multiple comparisons. This method helps control the family-wise error rate and offers detailed information on where the differences are.

The following section summarizes all pairwise comparisons between HbA1c quartiles (Q1–Q4) for each neurotransmitter in the PreDM and T2DM groups. Both significant and non-significant results are included, highlighting important trends (Table 6 and Figure 2).

DA—no pairwise comparisons between HbA1c quartiles reached statistical significance for DA, despite the overall Kruskal–Wallis test being significant (*p* = 0.047). This indicates that the overall trend (e.g., a gradual increase) may not be caused by sharp differences between any two quartiles, but rather by a general gradient.

Patients with moderately elevated HbA1c (Q3) had significantly higher NE levels than those in the Q2 range. This suggests that sympathetic nervous system activation (via NE) may become more pronounced in the 8.6–10.7% HbA1c range, potentially reflecting early autonomic dysregulation: significant difference between Q2 (HbA1c: 7.21–8.65) and Q3 (HbA1c: 8.66–10.73); mean difference: +4.737 pg/mL in Q3 versus Q2, 95% CI: (+0.075, +9.400), *p*-value = 0.045.

### 2.3. Regression Analyses

#### 2.3.1. Multiple Linear Regression (MLR)

MLR models were constructed for each neurotransmitter (DA, NE, EPI, and ST) using the following predictors: HbA1c, BMI, Age, TC, TG, TyG index, Sex, and Status (PreDM vs. T2DM). The goal was to evaluate the independent association between metabolic and lipid parameters and neurotransmitter levels. Predictors with *p* < 0.05 were considered statistically significant. A complete list of coefficient estimates and significance for each model is provided in Table 7.

The results of the MLR models suggest a consistent and robust association between T2DM status and elevated levels of all four neurotransmitters (DA: Coefficient β = +335.09, *p* = 0.0034; EPI: Coefficient β = +502.08, *p* < 0.001; NE: Coefficient β = +6.51, *p* < 0.001; ST: Coefficient β = +699.18, *p* < 0.001). These significant and consistent positive coefficients indicate a systemic neurochemical shift in T2DM compared to PreDM, independent of glycemia, lipids, or demographic factors.

The effect of sex (male) approached significance in the EPI model (EPI: Coefficient β = −127.11, *p* = 0.0624), with a negative coefficient indicating lower EPI levels in males compared to females. This is consistent with prior knowledge regarding sex-based differences in sympathetic nervous system activation and may indicate a biologically relevant, though statistically borderline, pattern.

Notably, HbA1c did not emerge as a significant independent predictor in any of the models. This suggests that while HbA1c reflects glycemic control, its explanatory power for neurotransmitter changes is likely overshadowed by the broader systemic differences between T2DM and PreDM captured by the binary disease status.

Finally, lipid-related markers (TC, TG, and TyG index) did not show statistically significant associations with any of the neurotransmitters. This suggests that isolated lipid abnormalities may not have a direct or measurable impact on neurochemical expression in this cohort, or that their influence is indirect and mediated through other metabolic pathways.

#### 2.3.2. Logistic Regression

A penalized logistic regression model (ridge regularization) was used to evaluate predictors of diabetes status (T2DM vs. PreDM), including neurotransmitter levels (DA, NE, EPI, ST), HbA1c, BMI, age, TC, TG, TyG index, and sex. All continuous predictors were standardized before model fitting.

The logistic model was fitted using penalized regression (ridge regularization) to minimize overfitting and account for potential multicollinearity between metabolic, lipid, and neurochemical predictors. All continuous variables were standardized (z-scored) prior to modeling, and a ridge-regularized model (α = 1.0) was fitted to minimize overfitting and control for multicollinearity. As such, the odds ratios (ORs) reflect the change in odds per 1 standard deviation increase in each predictor.

Table 8 summarizes the estimated logistic regression coefficients and odds ratios (OR) per 1 standard deviation (SD) for each predictor. Positive coefficients indicate increased odds of T2DM diagnosis, while negative coefficients suggest decreased odds.

Predictors with the most substantial impact:-ST showed the most considerable effect with an OR of 8.70, indicating that each 1 SD increase in ST level was associated with an almost 9-fold increase in the odds of T2DM;-NE and HbA1c were also strong predictors, with ORs of 3.76 and 3.15, respectively, suggesting meaningful contributions of both neurochemical activation and glycemic control.

Other positively associated predictors:-EPI and DA also increased T2DM odds (ORs: 2.61 and 1.38), supporting the idea that neurotransmitter elevations reflect disease severity;-Age contributed modestly (OR = 1.35), consistent with its established role as a risk factor for T2DM.

Predictors associated with decreased odds:-TC was the only variable with an OR clearly <1 (OR = 0.897), suggesting a potential inverse association, though the effect size was modest;-Variables like BMI, TG, TyG, and sex (male) were near OR = 1.00, implying minimal or no contribution to classification in this model when other variables are considered.

The discriminative performance of the model was evaluated using the area under the ROC curve (AUC). The AUC for this model was 1.000, indicating perfect separation between the T2DM and PreDM groups in this sample (Figure 3). This suggests strong discriminative power provided by the combined influence of glycemic, lipid, and neurochemical markers. Such a high AUC indicates that the model correctly classified all individuals in the sample with no false positives or false negatives. This perfect classification may be driven by the dominant effect of HbA1c, in combination with meaningful contributions from DA, NE, and TyG-related parameters. While the result confirms the strong internal validity of the extended logistic model, the unusually high AUC also raises the possibility of overfitting, particularly in relatively small or well-separated samples. External validation on an independent dataset is warranted to assess the generalizability of this predictive model. Nevertheless, these findings support the use of multidimensional predictors, including neurochemical biomarkers, for improving T2DM classification and early detection strategies.

### 2.4. Exploratory Correlation Analysis—Neurotransmitters and Metabolic Profile

In this study, we assessed the relationships between four neurotransmitters (DA, NE, EPI, and ST) and ten metabolic/lipid variables using Spearman correlation, resulting in 40 pairwise tests per group (PreDM and T2DM). To control for the risk of false positives due to multiple comparisons, we applied the Benjamini–Hochberg False Discovery Rate (FDR) correction.

The FDR method adjusts *p*-values to control the expected proportion of falsely rejected null hypotheses. Only correlations with an adjusted q-value < 0.05 were considered statistically robust. This approach retains statistical power while addressing the inflated Type I error rate common in high-dimensional comparisons.

#### 2.4.1. Exploratory Correlation Analysis in PreDM Group

Although no correlations passed the FDR threshold (q-value < 0.05), two pairwise associations in the PreDM group (Table 9) showed marginal significance at the unadjusted *p*-value level (<0.05). Specifically,

-EPI—HbA1c (rho = −0.348, *p*-value = 0.028, q-value (FDR) = 0.928): Suggests a potential inverse relationship between epinephrine and glycemic burden. This may reflect reduced adrenergic activity as glucose regulation worsens.-ST—TyG-BMI (rho = 0.312, *p*-value = 0.050, q-value (FDR) = 0.935): Indicates a possible link between serotonergic activation and worsening insulin resistance or visceral adiposity.

These findings should be interpreted cautiously and validated in larger samples. They offer hypothesis-generating insights for future investigation.

#### 2.4.2. Exploratory Correlation Analysis in the T2DM Group

Table 10 lists correlations that reached nominal significance (*p*-value < 0.05) but did not survive FDR correction (q-value ≥ 0.05). These are presented as exploratory signals.

These marginal correlations may highlight biologically plausible interactions, especially involving EPI and DA:-EPI—TyG-WC (rho = −0.365, *p*-value = 0.002, q-value (FDR) = 0.076): Suggests an inverse relationship between epinephrine levels and central insulin resistance markers.-EPI—TyG-WHtR (rho = −0.311, *p*-value = 0.009, q-value (FDR) = 0.158): Reinforces the above, linking epinephrine suppression with visceral metabolic burden.-DA—LDL-C (rho = 0.288, *p*-value = 0.016, q-value (FDR) = 0.159): Indicates potential coupling between dopaminergic activity and atherogenic lipid profiles.-DA—TC (rho = 0.290, *p*-value = 0.015, q-value (FDR) = 0.176): Indicates potential coupling between dopaminergic activity and atherogenic lipid profiles.-DA—HbA1c (rho = 0.269, *p*-value = 0.024, q-value (FDR) = 0.199): Dopamine increases in line with glycemic deterioration.

Although these findings did not reach statistical robustness after FDR adjustment, they may serve as hypotheses for future studies focused on neurochemical changes in the T2DM progression.

## 3. Discussion

Our study provides new evidence that systemic levels of neurotransmitters—DA, NE, EPI, and ST—are significantly higher in patients with newly diagnosed T2DM compared to PreDM individuals. This result remains consistent even after adjusting for factors such as glycemic control, lipid levels, age, BMI, and sex, suggesting that T2DM status itself is the primary independent predictor of neurotransmitter differences in our group. Additionally, in penalized logistic regression models, standardized levels of ST and NE proved to be strong predictors of T2DM status, outperforming traditional biomarkers such as HbA1c. Collectively, these findings suggest that neurochemical changes are a crucial, yet often underappreciated, aspect of early diabetes development. Below, we interpret the findings in relation to previous research, discuss possible mechanisms, acknowledge limitations, and suggest clinical implications and future research directions.

To our knowledge, few prior human studies have directly measured circulating neurotransmitter levels in PreDM or T2DM and linked them to metabolic or lipid profiles. Much of the existing literature focuses on central neurotransmission, brain imaging, or animal models, with limited application to peripheral biomarkers. For example, reviews have summarized the roles of catecholaminergic and serotonergic dysregulation in obesity, insulin resistance, and neuropsychiatric conditions [5], as well as serotonergic dysfunction in diabetes complications [15]. Similarly, the role of DA in glucose regulation has been reviewed, emphasizing its interactions in islet signaling and metabolism, mainly from experimental or central nervous system perspectives [16]. Our study provides unique translational insight by measuring multiple monoaminergic biomarkers in the serum of newly diagnosed T2DM and PreDM patients, along with detailed lipid and insulin resistance indices—a combination not previously reported. This marks a significant advance in identifying peripheral neurochemical signatures of early diabetes.

The T2DM status emerged as the only consistent and robust independent predictor across all neurotransmitter models, indicating that the transition from PreDM to overt diabetes involves systemic neurochemical reprogramming rather than a gradual correlation with glucose levels alone. The insignificance of HbA1c in these models highlights that glycemic control alone cannot account for differences in neurotransmitters when disease status and systemic factors are considered. This may suggest that once metabolic dysregulation reaches a certain threshold, neurochemical changes occur relatively quickly as part of a “diabetic milieu.”

In the logistic regression model, the strong effect sizes of ST (OR ≈ 8.70 per 1 SD) and NE (OR ≈ 3.76) highlight their potential as predictive biomarkers for differentiating T2DM from PreDM beyond traditional risk factors. The fact that ST—often seen as a key neuromodulator—is among the top predictors is notable and suggests possible peripheral functions or interactions with metabolic tissues (such as adipose tissue and gut). The smaller but still positive ORs for EPI, DA, and age indicate they contribute modestly but cumulatively to disease classification.

Predictors such as BMI, TG, TyG, and sex had ORs close to 1.00 in the fully adjusted model, indicating minimal independent contribution when neurochemical and glycemic markers are included. This suggests that neurochemical variation may reflect underlying metabolic dysregulation not fully captured by traditional anthropometric or lipid variables.

Our exploratory Spearman correlation analyses, although not all surviving FDR correction, revealed trends between neurotransmitter levels and composite insulin resistance indices (TyG-WC, TyG-WHtR) or lipoprotein ratios, especially within the T2DM subgroup. These associations, while preliminary, suggest that elevated NE and DA may be linked to worsening insulin resistance and dyslipidemia in diabetes. Although these findings did not survive FDR correction, they suggest a biologically plausible connection between metabolic and neurochemical pathways that may become more pronounced as the disease progresses.

The lack of strong, significant correlations in the PreDM group supports our view that neurochemical remodeling is emerging rather than already present. In PreDM, metabolic stress may not yet be enough to disrupt autonomic or serotonergic systems enough to cause visible systemic changes.

The observed rise in catecholamines and ST in T2DM is supported by mechanistic literature. Pancreatic β-cells and α-cells can synthesize dopamine and catecholamines, which act in an autocrine/paracrine fashion to modulate insulin secretion. Excessive local catecholamine tone may blunt insulin response and contribute to glycemic dysregulation [17]. Experimental studies demonstrate that DA (or its precursor L-dopa) can inhibit glucose-stimulated insulin secretion, possibly via suppression of β-cell excitability or by receptor-mediated feedback [18]. Peripheral ST signaling influences hepatic gluconeogenesis, adipocyte lipolysis, and insulin sensitivity. Dysregulation in serotonergic tone may exacerbate insulin resistance and contribute to dyslipidemia [19,20]. Bromocriptine, a dopaminergic agonist, is approved for the treatment of T2DM and acts partly by modulating central sympathetic tone and peripheral metabolism. This lends indirect translational support to the idea that manipulating monoamine pathways may alter metabolic outcomes [21]. Disruption of dopaminergic, serotonergic, and adrenergic pathways has been linked to insulin resistance, increased body fat, and altered stress responses, all key features of metabolic syndrome [22]. Additionally, the reciprocal relationship between diabetes, stress hormones, and emotional health indicates that neuroendocrine markers could improve metabolic risk assessment beyond traditional measures [23].

Therefore, our findings of elevated DA, NE, EPI, and ST levels are biologically plausible considering increasing metabolic stress, sympathetic activation, and peripheral monoamine imbalance.

From a practical standpoint, these results open several promising avenues:Biomarker development: ST, NE, and DA may function as supplementary biomarkers for early disease detection, risk assessment, or tracking disease progression, especially when used alongside glycemic and lipid measurements.Refining risk models: Incorporating neurochemical markers into multivariable risk prediction could improve accuracy in identifying individuals transitioning from PreDM to T2DM or at risk for complications.Therapeutic targeting: If further validated, monoamine pathways could become treatment targets. For example, dopaminergic modulation (such as in bromocriptine) or serotonergic modulation might affect metabolic outcomes beyond just glycemia.Patient stratification: Patients with unusually high neurotransmitter levels may constitute a distinct “neuro-metabolic phenotype” of T2DM, justifying customized preventive or therapeutic strategies.

We acknowledge several limitations:Cross-sectional design: Causality cannot be determined. Longitudinal studies are necessary to establish the directionality (i.e., whether changes in neurotransmitters precede or follow metabolic deterioration).Measurement context: Circulating levels of neurotransmitters may not accurately reflect tissue-specific or central concentrations; peripheral factors (such as clearance, transporter function, and degradation) can influence serum levels.Sample size and power: Although sufficient for primary comparisons, limited power may explain the lack of significance in some adjusted models or correlations.Assay specificity and stability: Quantifying neurotransmitters in serum or plasma can be affected by preanalytical variables such as stability and degradation, requiring rigorous standardization.Potential confounding factors such as diet, medications, or stress exposure were not fully controlled, which could affect neurotransmitter levels.Future research should incorporate prospective cohorts, intervention studies targeting monoamine pathways, simultaneous measurement of tissue and CSF levels, and integration with imaging or genetic biomarkers to map the neuro-metabolic axis thoroughly.

## 4. Materials and Methods

Over a period of six months, we conducted a non-interventional, exploratory epidemiological study. This study involved enrolling 190 consecutive patients with newly diagnosed T2DM and 100 patients with preDM who matched the inclusion criteria in terms of age, gender ratio, and urban/rural location, serving as the control group. The study adhered to the Declaration of Helsinki and received approval from the Ethics Committee of the Clinical Municipal Hospital Filantropia (no. 886/15 January 2024) and the Emergency County Clinical Hospital of Craiova (no. 2371/14 January 2022), Dolj, Romania.

### 4.1. Patient Selection

To be included in the study, certain conditions had to be met. We selected individuals with T2DM who were at least 18 years old from the Outpatient Diabetes, Nutrition, and Metabolic Diseases departments at Craiova Clinical Municipal Hospital and Emergency County Clinical Hospital. All participants were included in the study voluntarily after signing an informed consent form.

Patients with chronic microvascular complications of T2DM, such as diabetic peripheral polyneuropathy, diabetic kidney disease, and diabetic retinopathy, were not included in the study. Additionally, patients under 18, pregnant women, those with type 1 diabetes, and individuals who had experienced an acute infection or inflammatory disease in the past month were excluded. Patients with ongoing infections, inflammatory conditions, or cancer were also excluded from the study.

We studied 70 newly diagnosed T2DM and 40 preDM patients, based on several exclusion criteria (120 T2DM patients were lost to follow-up, with reasons including diabetic peripheral polyneuropathy (*n* = 40), diabetic kidney disease (*n* = 30), diabetic retinopathy (*n* = 30), unwillingness to continue (*n* = 10), and relocation (*n* = 10)).

### 4.2. Assessment of Diabetes and Prediabetes

There are several ways to define prediabetes, including: (1) a diagnosis made by a healthcare professional; (2) a hemoglobin A1c (HbA1c) level between 5.7% and less than 6.5%; (3) a fasting plasma glucose (FPG) level between 5.6 mmol/L and 7.0 mmol/L; or (4) a 2-h FPG value during an oral glucose tolerance test (OGTT) between 7.8 mmol/L and 11.0 mmol/L [24].

Diabetes is diagnosed when at least one of the following conditions is met: a medical diagnosis verified by the patient’s healthcare providers, an HbA1c level above 6.5%, an FPG level of 7.0 mmol/L or higher, a random blood glucose level of 11.1 mmol/L or higher, or a two-hour blood glucose level exceeding 11.1 mmol/L after an OGTT. Additionally, a random glucose value accompanied by classic hyperglycemic symptoms, such as polyuria, polydipsia, and unexplained weight loss, or hyperglycemic crises [24], may indicate diabetes.

### 4.3. Medical History, Biometric Parameter Evaluation, and Demographic Information

Information on physical measurements, medical conditions, lab test results, and personal and lifestyle factors was collected through an interview questionnaire.

Demographic factors included age, sex, monthly household income, and educational level. Lifestyle and health-related factors included a history of smoking and drinking, family history of conditions like hypertension, diabetes, and heart disease, and the amount of time spent on moderate physical activity each week.

### 4.4. Laboratory Investigations

After obtaining the anthropometric measurements, we conducted more in-depth assessments with the subjects in the laboratory setting.

#### 4.4.1. Sample Collection

As part of the biological sampling procedure, two blood samples from each patient were collected into two different tubes:-Two additive-free tubes (Becton Dickinson Vacutainer, Franklin Lakes, NJ, USA) with approximately 5 mL of venous blood from each patient. In line with standard processing protocols, blood samples were allowed to clot and then centrifuged within 4 h of collection at 3000× *g* for 10 min using a Hermle centrifuge (Hermle AG, Gosheim, Baden-Württemberg, Germany). The resulting serum from one tube was aliquoted into pre-labeled vials, sealed tightly to prevent contamination, and stored at controlled temperatures between −20 °C and −80 °C to ensure sample preservation. To maintain specimen integrity, freeze–thaw cycles were strictly avoided. Before analysis, frozen serum samples were passively thawed to reach room temperature. These aliquots were used for immunological investigations. The serum from the second tube was used for biochemical investigations.-And peripheral venous blood collected in ethylene-diamine-tetra-acetic acid (EDTA) vacutainer tubes (Becton Dickinson Vacutainer, Franklin Lakes, NJ, USA) was used to perform a complete blood count (CBC). Utilizing flow cytometry under Coulter’s principle, we successfully established an extended leukocyte differential by analyzing five distinct parameters using the MINDRAY BC-6800 (Mindray, Shenzhen, China). This approach enabled us to effectively identify and characterize a range of hemoleucogram markers.

#### 4.4.2. Biochemical Investigations

Biochemical parameters were measured using the ARCHITECT c4000 clinical chemistry analyzer (Abbott Laboratories, IL, USA), which employs photometric, enzymatic, and potentiometric methods depending on the analyte. Serum or plasma samples were analyzed according to standardized protocols provided by the manufacturer. All measurements were performed in accordance with the instrument’s calibration and quality control procedures.

FPG and 2hPG were determined using an enzymatic hexokinase/glucose-6-phosphate dehydrogenase (G6PDH) method. The reaction produces NADPH, which is measured photometrically at 340 nm.TC was measured by an enzymatic colorimetric method involving cholesterol esterase and cholesterol oxidase, with quantification based on the generation of hydrogen peroxide (H_2_O_2_) and subsequent chromogenic reaction.TG were assessed enzymatically using lipoprotein lipase, glycerol kinase, and glycerol-3-phosphate oxidase, resulting in H_2_O_2_ formation, which was detected photometrically.HDL-C and LDL-C were measured using direct enzymatic methods with selective inhibition or solubilization, allowing specific quantification of HDL-C or LDL-C, respectively, without prior sample pretreatment.CREA was determined using a specific enzymatic method, based on the conversion of creatinine to sarcosine, followed by oxidation and H_2_O_2_ generation, providing improved accuracy over the traditional Jaffe method.HbA1c was measured using an enzymatic photometric assay. Hemoglobin was lysed, and the N-terminal valine of HbA1c was selectively cleaved and oxidized by fructosyl-peptide oxidase, forming H_2_O_2_, which was then quantified. HbA1c results were expressed as a percentage (%) of total hemoglobin.

Serum creatinine concentrations were assessed, and the estimated glomerular filtration rate (eGFR) was determined using the Modification of Diet in Renal Disease (MDRD) formula [25].

### 4.5. Determination of TyG and TyG-Related Indices

The formulas used to compute TyG and TyG-derived indices are as follows [26,27]:TyG  =  ln [triglycerides (mg/dL)  ×  glucose (mg/dL)/2];(1)TyG-WC  =  TyG  ×  waist circumference;(2)TyG-WHtR  =  TyG  ×  waist to height ratio;(3)TyG-BMI  =  TyG  ×  BMI.(4)

### 4.6. Immunological Assessment

Researchers at the Immunology Laboratory of the University of Medicine and Pharmacy of Craiova used the Enzyme-Linked Immunosorbent Assay (ELISA) technique to measure the serum levels of DA, ST, NE, and EPI.

#### 4.6.1. Assays and Specificity

Serum concentrations of DA, NE, EPI, and ST were measured using ELISA kits from Elabscience (Houston, TX, USA). See Table 11 for details. According to the manufacturer’s Instructions for Use (IFU) and lot-specific validation sheets, all antibodies used in these assays were raised against hapten-conjugated targets and tested for cross-reactivity with structurally related catecholamines and metabolites. The reported cross-reactivity for each analyte: No significant cross-reactivity or interference was observed between Universal DA (https://789.bio/ea/0mPyrL (accessed on 25 September 2025)), Universal NE (https://789.bio/ea/Gy9CmL (accessed on 25 September 2025)), Universal EPI (https://789.bio/ea/rDqDmD (accessed on 25 September 2025)), Universal ST (https://789.bio/ea/frTSy1 (accessed on 25 September 2025)), and their analogs.

Cross-reactivity data for catecholamine antibodies are rarely published. In a competitive dopamine ELISA, Kim et al. [28] reported 18.9% cross-reactivity with epinephrine and 3-methoxytyramine, while other catecholamines such as norepinephrine or 3,4-dihydroxyphenylacetic acid (DOPAC) showed <1% cross-reactivity. Comparable commercial assays (Weldon Biotech CAT-E-75e2) indicate cross-reactivity values between adrenaline/noradrenaline/dopamine antibodies and catecholamine analogs: for example, an <0.030% cross-reactivity for norepinephrine with adrenaline antibody, an <0.020% cross-reactivity for adrenaline with dopamine antibody, and an <0.012% cross-reactivity for adrenaline with norepinephrine antibody [29]. A review on “Changing Cross-Reactivity for Different Immunoassays” [30] emphasizes that cross-reactivity is not only a property of the antibody, but also depends on the sample format, antibody/antigen concentrations, and the mode (equilibrium vs. kinetic) of the immune reaction. These studies confirm the high specificity but partial overlap inherent to anti-catecholamine antibodies. We have incorporated these references in the revised manuscript to support the specificity of our immunoassays. Due to proprietary constraints, we did not independently re-evaluate all cross-reactivities in our lab.

#### 4.6.2. Pre-Analytical Handling

To preserve analyte stability and reduce the risk of non-specific oxidation or degradation, all samples were processed within 30 min of collection. After clotting and centrifugation, serum was aliquoted into cryovials and stored at −80 °C. Each sample was thawed only once before analysis. Where applicable, manufacturer-supplied diluents containing stabilizing agents were used to minimize assay interference.

#### 4.6.3. Analytical Limitations and Quality Controls

In-house spike-and-recovery or parallelism experiments were not performed. Therefore, although manufacturer documentation indicates high specificity and no significant cross-reactivity, analytical interference cannot be entirely ruled out. Additionally, orthogonal validation via liquid chromatography coupled with mass spectrometry (LC-MS/MS) is planned for a subset of samples in future work.

#### 4.6.4. Test Principle

This ELISA kit employs the competitive-ELISA principle. The micro ELISA plate included in the kit has been pre-coated with Universal DA/NE/EPI/ST. During the reaction, Universal DA/NE/EPI/ST in samples or standards competes with a fixed amount of Universal DA/NE/EPI/ST on the solid-phase support for binding sites on the biotinylated detection antibody specific to Universal DA/NE/EPI/ST. Excess conjugate, along with unbound samples or standards, is washed away from the plate. Avidin conjugated to horseradish peroxidase (HRP) is then added to each well and incubated. Next, a TMB substrate solution is added to each well. The addition of stop solution halts the enzyme-substrate reaction, and the color change is measured spectrophotometrically at a wavelength of 450 ± 2 nm, using the Asys Expert Plus UV G020 150 Microplate Reader from ASYS Hitech GmbH, Eugendorf, Austria. The concentration of Universal DA/NE/EPI/ST in the samples is determined by comparing the optical density (OD) of the samples to the standard curve. In Table 12, we have included the calibration curves for each ELISA experiment.

### 4.7. Statistical Analysis

Using Microsoft Excel, we processed and managed patient data from medical records. To analyze the data, we used GraphPad Prism 10.6.1 (GraphPad Software, LLC, San Diego, CA, USA). We applied the Kolmogorov–Smirnov and Shapiro–Wilk tests to determine if the data were normally distributed.

The data following a normal distribution were analyzed statistically using Welch’s *t*-test. Data from Table 3 and Table 4 were examined with the two-way ANOVA (Kruskal–Wallis) test, and data from Table 5 were examined with the one-way ANOVA (Kruskal–Wallis) test. Additionally, categorical variables are reported as percentages.

For comparisons across HbA1c quartiles (Q1–Q4), one-way ANOVA was applied separately to each neurotransmitter. When the overall ANOVA was statistically significant (*p* < 0.05), post hoc analysis using the Tukey Honestly Significant Difference (HSD) test was conducted to determine specific pairwise differences between quartile groups.

Multivariate statistical analyses were conducted to evaluate the associations between neurotransmitter levels (DA, NE, EPI, ST) and metabolic, anthropometric, and demographic variables. Multiple linear regression (MLR) models were constructed for each neurotransmitter individually, using HbA1c, BMI, Age, TC, TG, TyG index, sex, and Status (PreDM vs. T2DM) as predictors. Model assumptions were verified by visual inspection of residual plots to confirm normality and homoscedasticity, and variance inflation factors (VIF < 2) were computed to exclude multicollinearity. Influential observations were assessed using standardized residuals and Cook’s distance (threshold 4/n). In addition, a penalized logistic regression model was used to evaluate whether neurotransmitter levels, in conjunction with HbA1c, BMI, Age, TC, TG, TyG index, and sex, could predict diabetes status (T2DM vs. PreDM). Predictors were standardized (z-scores), and a ridge-regularized model (alpha = 1.0) was fitted to minimize overfitting and control for multicollinearity. The discriminative performance of the model was evaluated using the area under the ROC curve (AUC). Statistical significance was set at *p* < 0.05.

Pairwise correlations between neurotransmitter concentrations (DA, NE, EPI, ST) and metabolic or lipid variables (HbA1c, TC, TG, HDL-C, LDL-C, TG/HDL-C ratio, TyG index, TyG-BMI, TyG-WC, TyG-WHtR) were assessed using Spearman’s rank correlation coefficient (rho) separately in the PreDM and T2DM subgroups. The normality of continuous variables was not assumed. For each pairwise comparison, two-tailed *p*-values were calculated. To control for multiple comparisons (40 correlations per group), a False Discovery Rate (FDR) correction was applied using the Benjamini–Hochberg procedure in GraphPad Prism 10.6.1(892). The resulting q-values represent adjusted *p*-values that reflect the expected proportion of false positives among the rejected hypotheses. A correlation was considered statistically significant if q < 0.05. Correlations with *p* < 0.05 but q ≥ 0.05 were reported as exploratory trends, requiring further validation and confirmation. Results were summarized in tabular form, and significant correlations were highlighted for each metabolic subgroup.

## 5. Conclusions

In conclusion, our study provides strong evidence that newly diagnosed T2DM is associated with systemic elevations of multiple neurotransmitters, independent of glycemic and lipid measures. The dominance of T2DM status as a predictor and the discriminative power of ST and NE in logistic models underscore the potential of peripheral neurochemical profiling as a novel dimension in diabetes research. This work lays the foundation for future translational studies to elucidate the pathological roles of monoamines in metabolic disease and to explore therapeutic targeting of neurochemical pathways in diabetes management.

## Figures and Tables

**Figure 1 ijms-26-10068-f001:**
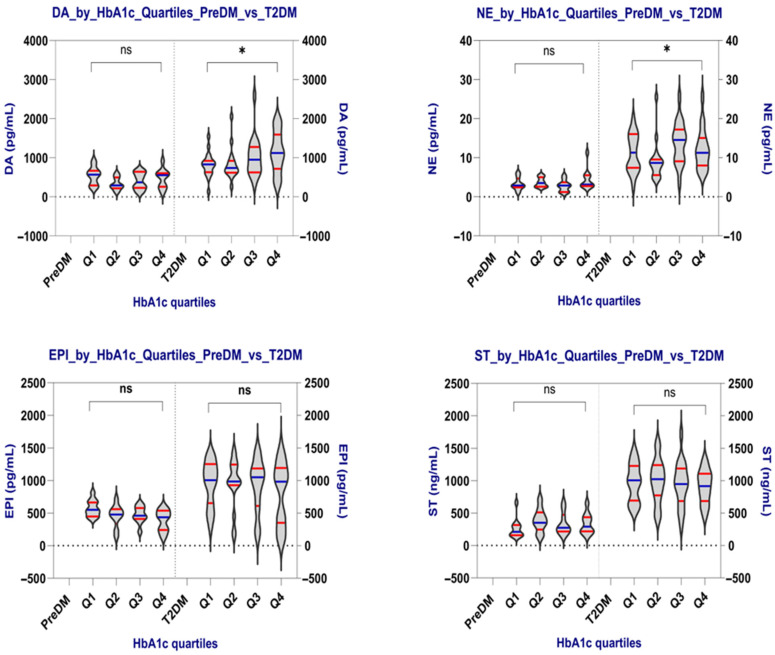
Neurotransmitter levels by HbA1c quartiles in the PreDM and T2DM groups. The violin plots display the distribution of dopamine (DA), norepinephrine (NE), epinephrine (EPI), and serotonin (ST) across HbA1c quartiles (Q1–Q4) in both groups. Horizontal blue lines represent medians, while horizontal red lines indicate the quartiles. Values are shown in pg/mL (DA, NE, EPI) and ng/mL (ST). Statistical differences were tested with the one-way ANOVA (Kruskal–Wallis) test. DA and NE exhibited statistically significant differences between quartiles (*p* = 0.047 and *p* = 0.032, respectively), indicating progressive changes with higher glycemic burden. EPI and ST levels remained stable; * *p* < 0.05: statistically significant; ns: not significant.

**Figure 2 ijms-26-10068-f002:**
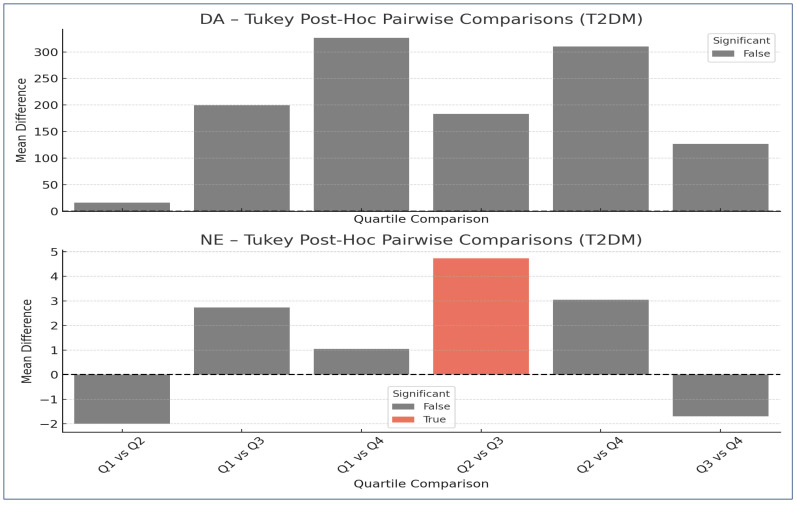
Tukey HSD Post hoc Pairwise Comparisons for Dopamine (DA) and Norepinephrine (NE) Across HbA1c Quartiles in T2DM Patients. This figure illustrates the mean differences in neurotransmitter levels (dopamine (DA), and norepinephrine (NE)) between HbA1c quartiles in the T2DM group, based on Tukey’s Honestly Significant Difference (HSD) post hoc test following Kruskal–Wallis test. Each bar represents the mean difference in concentration between two quartiles. Positive values indicate higher neurotransmitter levels in the second quartile of the comparison, negative values indicate higher levels in the first quartile. Color coding: red bars represent comparisons that reached statistical significance (*p* < 0.05), gray bars represent non-significant differences (*p* ≥ 0.05). A dashed horizontal line at y = 0 indicates no difference between groups. Notably, a significant increase in NE was observed between Q2 and Q3, indicating enhanced sympathetic activity in the presence of moderate hyperglycemia. No pairwise comparisons reached significance for DA despite the overall trend being significant.

**Figure 3 ijms-26-10068-f003:**
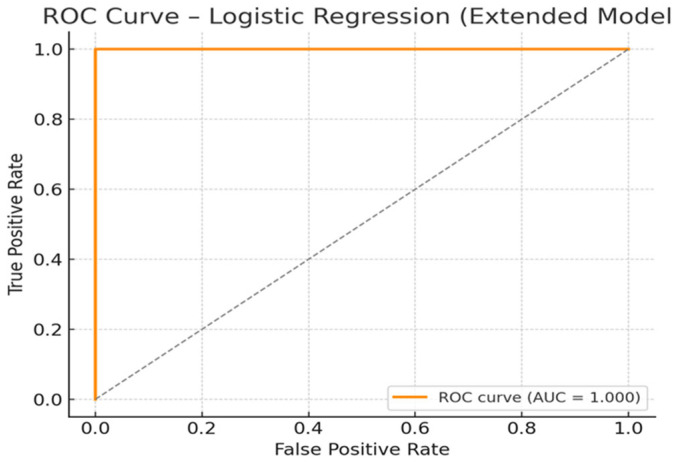
Receiver Operating Characteristic (ROC) Curve for the penalized logistic model incorporating neurotransmitter and metabolic predictors, used to classify T2DM vs. PreDM status. The orange curve represents the model’s true positive rate (sensitivity) versus false positive rate (1–specificity) across all thresholds, with an area under the curve (AUC) of 1.000, indicating perfect discrimination in this dataset. The diagonal dashed line corresponds to a random classification (AUC = 0.5).

**Table 1 ijms-26-10068-t001:** Clinical and Demographic Characteristics of Patients with PreDM and T2DM.

Parameters	PreDM Group (*n* = 40)	T2DM Group (*n* = 70)	*p*-Value
Age (yrs) (mean ± SD)	56.18 ± 12.09	63.87 ± 11.02	0.002
Gender, Female/Male (*n*)	18/22	33/37	1.000
Area of residence, Rural/Urban (*n*)	19/21	30/40	0.786
Hypertension, *n* (%)	34 (85.00%)	47 (67.14%)	0.069
Dyslipidemia, *n* (%)	31 (77.50%)	47 (67.14%)	0.351
Hepatosteatosis, *n* (%)	26 (65.00%)	51 (72.86%)	0.516
SBP (mmHg) (mean ± SD)	134.32 ± 16.50	135.00 ± 16.12	0.792
DBP (mmHg) (mean ± SD)	79.51 ± 13.04	78.91 ± 13.10	0.812
Anthropometric characteristics			
BMI (kg/m^2^) (mean ± SD)	30.57 ± 8.41	29.32 ± 4.47	0.384
BMI category (*n*)			
Normal (18.5–24.9 kg/m^2^)	14	18	
Overweight (25–29.9 kg/m^2^)	11	25	
Obese (≥30 kg/m^2^)	15	27	
Height (m) [median (range)]	1.60 (1.5–1.88)	1.65 (1.48–1.89)	0.222
Weight (kg) [median (range)]	73.25 (50.0–140.5)	79.00 (62.0–137.0)	0.285
WC (cm) (mean ± SD)	97.25 ± 14.35	96.11 ± 15.54	0.576
HC (cm) [median (range)]	105.50 (82.0–147.0)	106.00 (89.0–141.0)	0.893
WHR (mean ± SD)	0.90 ± 0.08	0.90 ± 0.12	0.791
WHtR (mean ± SD)	0.61 ± 0.09	0.58 ± 0.09	0.188
Glycemic metabolism			
FPG (mg/dL) [median (range)]	103.50 (56.0–121.0)	143.00 (106.0–273.0)	<0.0001
2hPG (mg/dL) [median (range)]	151.50 (141.0–196.0)	248.00 (129.0–475.0)	<0.0001
HbA1c (%) (mean ± SD)	5.97 ± 0.22	9.04 ± 2.16	<0.0001
Lipid profile and derived indices			
TC (mg/dL) [median (range)]	177.50 (94.0–377.0)	199.00 (84.0–464.57)	0.213
TG (mg/dL) [median (range)]	118.00 (44.0–387.0)	118.50 (45.0–345.0)	0.986
TyG index [median (range)]	8.64 (7.73–9.89)	8.97 (7.84–10.77)	0.001
TyG-BMI (mean ± SD)	266.40 ± 77.56	268.60 ± 49.31	0.128
TyG-WC (mean ± SD)	840.80± 123.00	878.10 ± 153.40	0.038
TyG-WHtR (mean ± SD)	5.23 ± 0.76	5.30 ± 0.92	0.102
TG/HDL-c [median (range)]	2.42 (0.59–5.56)	2.88 (0.65–15.22)	0.117
LDL-C (mg/dL) [median (range)]	113.10 (35.60–368.00)	96.80 (41.00–296.00)	0.434
HDL-C (mg/dL) (mean ± SD)	51.22 ± 12.42	45.03 ± 14.978	0.022
Renal and hematological function			
eGFR (mL/min/1.73 m^2^) MDRD (mean ± SD)	70.96 ± 26.08	93.52 ± 30.43	<0.0001
CREA [median (range)]	0.87 (0.56–1.75)	0.74 (0.42–2.35)	0.0557 *
Hb (g/dL) (mean ± SD)	13.38 ± 2.05	13.18 ± 1.92	0.520
WBC (×10^3^/μL) [median (range)]	7.21 (5.27–12.66)	8.62 (4.21–25.36)	0.0577 *

SBP: systolic blood pressure; DBP: diastolic blood pressure; WC: Waist circumference; HC: Hip circumference; WHR: waist to hip ratio; WHtR: waist to height ratio; BMI: body mass index; FPG: fasting plasma glucose; 2hPG: two-hour plasma glucose after a 75 g oral glucose tolerance test; HbA1c: glycosylated hemoglobin A1c; TC: total cholesterol; TG: total triglycerides; LDL-C: low-density lipoprotein cholesterol; HDL-C: high-density lipoprotein cholesterol; eGFR: estimated glomerular filtration rate; MDRD-Study: modification of diet in renal disease study; CREA: creatinine; Hb: haemoglobin; WBC: white blood cells/leukocytes; TyG: triglyceride-glucose index; TyG-BMI: TyG adjusted for BMI; TyG-WC: TyG adjusted for WC; TyG-WHtR: TyG adjusted for WHtR; SD: standard deviation; the normal distribution data were statistically analyzed using the GraphPad Prism 10.6.1 data analysis tool, applying the Welch’s *t* test; the nonparametric data were statistically analyzed using the two-way ANOVA (Kruskal–Wallis) test; the Chi-square test was used for categorical data; *: reached the significance limit.

**Table 2 ijms-26-10068-t002:** Comparing the Neurotransmitter Values between the PreDM and T2DM groups.

Parameter	PreDM (*n* = 40)	T2DM (*n* = 70)	*p*-Value
DA (pg/mL) mean ± SD median (range)	445.17 ± 222.45 461.74 (110.07–930.39)	955.10 ± 458.32 865.20 (135.68–2587.00)	<0.0001
ST (ng/mL) mean ± SD median (range)	327.24 ± 164.73 285.70 (142.90–713.40)	957.86 ± 305.84 987.30 (297.40–1720.00)	<0.0001
NE (pg/mL) mean ± SD median (range)	3.56 ± 1.87 2.93 (1.07–11.33)	11.70 ± 5.32 10.88 (2.73–25.53)	<0.0001
EPI (pg/mL) mean ± SD median (range)	467.17 ± 157.43 476.90 (127.60–816.10)	910.23 ± 400.43 991.00 (121.40–1482.00)	<0.0001

DA: dopamine; ST: serotonin; EPI: adrenaline/epinephrine; NE: noradrenaline/norepinephrine; the nonparametric data were statistically analyzed using the GraphPad Prism 10.6.1 data analysis tool, applying the two-way ANOVA (Kruskal–Wallis) test.

**Table 3 ijms-26-10068-t003:** Comparing neurotransmitter levels according to the BMI categories in the PreDM and T2DM groups.

Parameters		PreDM			T2DM			
	Normal (*n* = 14)	Overweight (*n* = 11)	Obese (*n* = 15)	*p*	Normal (*n* = 18)	Overweight (*n* = 25)	Obese (*n* = 27)	*p*
DA (pg/mL) mean ± SD median (range)	497.30 ± 226.66	429.88 ± 238.90	407.72 ± 212.06	0.387	956.84 ± 458.29	902.18 ± 457.69	1002.90 ± 470.84	0.814
	557.84 121.97–20.12	456.80 110.07–930.39	295.62 129.25–671.60		900.28 469.61–2587.14	765.65 135.68–1977.70	985.21 262.47–2059.50	
NE (pg/mL) mean ± SD median (range)	3.70 ± 1.53	3.89 ± 2.81	3.20 ± 1.30	0.797	12.45 ± 5.24	10.67 ± 4.23	12.03 ± 6.26	0.481
	3.12 1.07–6.22	2.867 1.14–11.33	2.91 1.21–5.45		11.30 4.85–25.51	10.80 4.35–20.00	9.62 2.73–25.53	
EPI (pg/mL) mean ± SD median (range)	453.33 ± 158.77	459.05 ± 204.37	510.05 ± 117.53	0.406	947.04 ± 412.02	898.41 ± 441.75	896.64 ± 364.88	0.792
	463.98 127.61–64.29	446.69 142.89–816.07	507.10 212.08–706.63		1010.20 121.38–1437.20	987.26 127.21–1482.40	962.78 132.84–1447.20	
ST (ng/mL) mean ± SD median (range)	279.46 ± 147.26	296.95 ± 176.25	394.06 ± 159.63	0.078	939.11 ± 338.42	949.73 ± 334.26	977.87 ± 263.63	0.772
	225.30 142.89–53.57	229.30 149.12–713.38	344.64 226.05–664.29		968.15 444.64–1719.80	967.62 297.41–1573.30	994.97 444.64–1377.70	

DA: dopamine; ST: serotonin; EPI: adrenaline/epinephrine; NE: noradrenaline/norepinephrine; data were statistically analyzed using the GraphPad Prism 10.6.1, applying the one-way ANOVA (Kruskal–Wallis) test.

**Table 4 ijms-26-10068-t004:** Comparing neurotransmitter levels according to Gender in the PreDM and T2DM groups.

Parameters			PreDM		T2DM		
		Male (*n* = 18)	Female (*n =* 22)	*p*-Value	Male (*n* = 37)	Female (*n* = 33)	*p*-Value
DA (pg/mL)	mean ± SD	412.09 ± 207.05	472.23 ± 235.56	0.487	927.11 ± 518.71	986.49 ± 385.11	0.749
	median (range)	295.62 134.47–834.31	501.47 110.07–930.39		839.22 135.68–2587.40	893.49 469.61–1977.10	
NE (pg/mL)	mean ± SD	3.36 ± 1.44	3.73 ± 2.18	0.768	11.98 ± 5.20	11.37 ± 5.51	0.496
	median (range)	2.92 1.07–6.22	2.95 1.21–11.33		11.27 4.35–25.53	9.00 2.73–25.53	
EPI (pg/mL)	mean ± SD	490.76 ± 112.41	464.23 ± 188.32	0.579	826.17 ± 419.83	1004.48 ± 360.72	0.072
	median (range)	491.35 199.60–706.63	474.07 127.61–816.07		984.92 121.38–1467.30	1070.40 132.84–1482.40	
ST (ng/mL)	mean ± SD	364.55 ± 183.49	296.72 ± 144.82	0.156	958.27 ± 305.04	957.39 ± 311.48	0.862
	median (range)	330.76 145.90–713.38	238.52 142.89–664.29		994.97 444.64–1522.30	949.04 297.41–1719.80	

DA: dopamine; ST: serotonin; EPI: adrenaline/epinephrine; NE: noradrenaline/norepinephrine; data were statistically analyzed using the GraphPad Prism 10.6.1, applying the two-way ANOVA (Kruskal–Wallis) test.

**Table 5 ijms-26-10068-t005:** Comparing neurotransmitter levels according to the HbA1c quartiles in the PreDM and T2DM groups.

PreDM Patients				
HbA1c Quartiles	Parameters			
	DA (pg/mL)	NE (pg/mL)	EPI (pg/mL)	ST (ng/mL)
Q1 (5.70–5.80) (*n* = 11) mean ± SD median (range)	543.20 ± 230.31	3.30 ± 1.60	572.20 ± 126.40	261.54 ± 151.04
	556.67 227.49–930.39	2.89 1.07–6.22	550.86 417.87–816.07	212.08 145.90–662.42
Q2 (5.81–5.90) (*n* = 9) mean ± SD median (range)	335.93 ± 160.76	3.69 ± 1.22	457.48 ± 173.09	390.62 ± 179.44
	294.40 121.97–600.59	3.54 2.48–5.84	478.39 142.89–770.38	351.80 153.56–713.38
Q3 (5.91–6.15) (*n* = 10) mean ± SD median (range)	406.38 ± 215.58	2.80 ± 1.44	474.48 ± 129.50	333.13 ± 170.90
	373.39 129.25–671.60	2.88 1.14–5.45	461.82 212.08–636.02	273.70 158.96–664.29
Q4 (6.16–6.45) (*n* = 10) mean ± SD median (range)	474.43 ± 244.42	4.50 ± 2.69	388.94 ± 161.75	336.58 ± 158.07
	557.84 110.07–920.12	3.14 2.65–11.33	435.88 127.61–597.98	291.58 149.12–653.57
*p*-value	0.240	0.398	0.125	0.250
**T2DM patients**				
Q1 (6.51–7.20) (*n* = 19) mean ± SD median (range)	820.65 ± 302.52	11.20 ± 4.86	952.70 ± 349.93	972.89 ± 295.44
	829.41 135.68–1547.30	11.33 2.73–20.00	1006.40 273.95–1412.10	942.21 535.50–1477.70
Q2 (7.21–8.65) (*n* = 16) mean ± SD median (range)	837.22 ± 412.00	9.19 ± 5.12	961.40 ± 381.57	1000.30 ± 336.12
	739.45 135.68–1547.30	8.67 2.73–20.00	987.26 273.95–1412.10	1023.00 535.50–1477.70
Q3 (8.66–10.73) (*n* = 18) mean ± SD median (range)	1020.30 ± 509.67	13.93 ± 5.32	922.21 ± 405.60	946.24 ± 346.67
	952.67 323.91–2587.40	14.58 3.73–25.51	1050.40 139.07–1447.20	949.04 297.41–1719.80
Q4 (10.74–15.50) (*n* = 17) mean ± SD median (range)	1147.30 ± 537.01	12.24 ± 5.29	801.92 ± 474.38	913.42 ± 259.38
	1124.30 270.38–1977.70	11.27 6.10–25.53	984.92 121.38–1482.40	917.20 474.38–1344.00
*p*-value	0.047	0.032	0.796	0.877

DA: dopamine; ST: serotonin; EPI: adrenaline/epinephrine; NE: noradrenaline/norepinephrine; data were statistically analyzed using the GraphPad Prism 10.6.1, applying the one-way ANOVA (Kruskal–Wallis) test.

**Table 6 ijms-26-10068-t006:** Tukey HSD (Honestly Significant Difference) post hoc pairwise comparisons of Dopamine (DA) and Norepinephrine (NE) across HbA1c quartiles in T2DM patients.

Neurotransmitter (pg/mL)	Group 1	Group 2	Mean Difference	*p*-Value	95% CI Lower	95% CI Upper	Significant
DA	Q1	Q2	16.568	0.999	−383.742	416.878	FALSE
DA	Q1	Q3	199.639	0.531	−188.411	587.688	FALSE
DA	Q1	Q4	326.635	0.138	−67.232	720.502	FALSE
DA	Q2	Q3	183.071	0.635	−222.291	588.432	FALSE
DA	Q2	Q4	310.067	0.203	−100.867	721.001	FALSE
DA	Q3	Q4	126.996	0.836	−272.004	525.996	FALSE
NE	Q1	Q2	−2.003	0.662	−6.607	2.601	FALSE
NE	Q1	Q3	2.735	0.378	−1.729	7.198	FALSE
NE	Q1	Q4	1.046	0.929	−3.484	5.576	FALSE
NE	Q2	Q3	4.737	0.045	0.075	9.400	TRUE
NE	Q2	Q4	3.049	0.332	−1.678	7.775	FALSE
NE	Q3	Q4	−1.689	0.767	−6.278	2.901	FALSE

The table presents the results of pairwise comparisons between HbA1c quartiles (Q1–Q4) for DA and NE levels in the T2DM group, following a statistically significant Kruskal–Wallis test. *Interpretation*: only one pairwise comparison was statistically significant. NE levels in Q3 were significantly higher than in Q2 (mean difference = +4.74 pg/mL, *p* = 0.045), suggesting enhanced sympathetic activation in patients with moderately elevated HbA1c (8.66–10.73%). Group 1 vs. Group 2: HbA1c quartile pairs being compared; Mean Difference: estimated difference in neurotransmitter levels between the two groups; *p*-value: adjusted *p*-value using the Tukey HSD method; 95% CI: Confidence interval of the mean difference; Significant: showing whether the comparison is statistically significant (*p* < 0.05).

**Table 7 ijms-26-10068-t007:** Summary table of Multiple Linear Regression (MLR) Coefficients estimates and significance for each model.

Neurotransmitter	Predictor	Coefficient β	Standard Error	t-Value	*p*-Value
DA	C (Sex) [T.Male]	−24.44	76.644	−0.319	0.7505
DA	C (Status) [T.T2DM]	335.09	111.618	3.002	0.0034 *
DA	HbA1c	48.55	30.832	1.575	0.1185
DA	BMI	2.10	6.204	0.338	0.7363
DA	Age	4.77	3.386	1.409	0.1618
DA	TC	0.90	0.594	1.517	0.1324
DA	TG	0.08	1.423	0.057	0.9544
DA	TyG	4.76	197.190	0.024	0.9808
EPI	C (Sex) [T.Male]	−127.11	67.453	−1.884	0.0624 **
EPI	C (Status) [T.T2DM]	502.08	98.234	5.111	<0.001 *
EPI	HbA1c	−26.68	27.135	−0.983	0.3278
EPI	BMI	1.05	5.460	0.193	0.8473
EPI	Age	1.76	2.980	0.592	0.5552
EPI	TC	−0.21	0.523	−0.399	0.6904
EPI	TG	0.47	1.253	0.373	0.7100
EPI	TyG	−24.72	173.544	−0.142	0.8870
NE	C (Sex) [T.Male]	0.38	0.895	0.426	0.6707
NE	C (Status) [T.T2DM]	6.51	1.303	4.997	<0.001 *
NE	HbA1c	0.58	0.360	1.610	0.1106
NE	BMI	−0.01	0.072	−0.109	0.9136
NE	Age	0.03	0.040	0.868	0.3876
NE	TC	0.00	0.007	0.583	0.5609
NE	TG	−0.01	0.017	−0.416	0.6780
NE	TyG	−0.54	2.302	−0.234	0.8152
ST	C (Sex) [T.Male]	−15.82	54.256	−0.291	0.7713
ST	C (Status) [T.T2DM]	699.18	79.014	8.849	<0.001 *
ST	HbA1c	−12.76	21.826	−0.585	0.5601
ST	BMI	4.57	4.392	1.042	0.3001
ST	Age	−1.32	2.397	−0.551	0.5831
ST	TC	0.16	0.420	0.381	0.7042
ST	TG	0.05	1.008	0.051	0.9594
ST	TyG	−27.03	139.589	−0.194	0.8469

The table summarizes the coefficient estimates and statistical significance for each predictor in the extended multiple linear regression (MLR) models constructed for the four neurotransmitters: dopamine (DA), norepinephrine (NE), epinephrine (EPI), and serotonin (ST). Each row in the table represents the result of a predictor from the model: NT ~ HbA1c + BMI + Age + TC + TG + TyG + Sex + Status (PreDM vs. T2DM). In summary, T2DM status remains the most dominant and consistent predictor of increased neurotransmitter levels, particularly in comparison to classical metabolic predictors. This finding highlights the significance of disease context and suggests that neuroendocrine alterations in diabetes are not fully accounted for by traditional metabolic markers alone. * *p* < 0.05: statistical significance; ** approached significance.

**Table 8 ijms-26-10068-t008:** Standardized Coefficients and Odds Ratios from Penalized Logistic Regression Model for Predicting T2DM Status.

Predictor	Coefficient (β)	OR Per 1 SD	Direction of Effect
z_ST	2.164	8.703	Increased odds
z_NE	1.325	3.763	Increased odds
z_HbA1c	1.146	3.145	Increased odds
z_EPI	0.958	2.607	Increased odds
z_DA	0.322	1.380	Increased odds
z_Age	0.301	1.351	Increased odds
z_BMI	<0.001	1.000	No effect
z_TG	<0.001	1.000	No effect
z_TyG	<0.001	1.000	No effect
Sex_Male	<0.001	1.000	No effect
z_TC	−0.108	0.897	Decreased odds

Neurotransmitters (especially ST and NE) were among the most powerful discriminators of T2DM status in this model. The model emphasizes that, beyond traditional metabolic indicators like HbA1c, neurochemical markers carry independent predictive value, reinforcing the emerging concept of neuro-metabolic dysregulation in diabetes.

**Table 9 ijms-26-10068-t009:** Exploratory Spearman correlations between neurotransmitters and metabolic/lipid variables in the PreDM Group.

Neurotransmitter	Variable	Spearman-Rho	*p*-Value	N	q-Value
DA	HbA1c	−0.134	0.409	40	0.999
DA	TC	−0.091	0.575	40	1.000
DA	TG	0.100	0.538	40	1.000
DA	TyG	0.080	0.622	40	1.000
DA	TyG-BMI	−0.048	0.768	40	1.000
DA	TyG-WC	0.000	0.999	40	1.000
DA	TyG-WHtR	0.092	0.570	40	1.000
DA	TG/HDL-C	0.138	0.396	40	0.997
DA	LDL-C	−0.055	0.736	40	1.000
DA	HDL-C	0.062	0.702	40	1.000
NE	HbA1c	0.137	0.398	40	0.999
NE	TC	−0.024	0.882	40	1.000
NE	TG	−0.139	0.393	40	0.995
NE	TyG	−0.005	0.974	40	1.000
NE	TyG-BMI	−0.129	0.428	40	1.000
NE	TyG-WC	0.141	0.386	40	0.970
NE	TyG-WHtR	0.085	0.602	40	1.000
NE	TG/HDL-C	−0.065	0.692	40	1.000
NE	LDL-C	0.006	0.971	40	1.000
NE	HDL-C	0.021	0.898	40	1.000
**EPI**	**HbA1c**	**−0.348**	**0.028**	**40**	**0.928**
EPI	TC	−0.036	0.825	40	1.000
EPI	TG	−0.166	0.307	40	0.957
EPI	TyG	−0.173	0.286	40	0.954
EPI	TyG-BMI	0.174	0.282	40	0.949
EPI	TyG-WC	−0.110	0.498	40	1.000
EPI	TyG-WHtR	−0.047	0.771	40	1.000
EPI	TG/HDL-C	−0.074	0.650	40	1.000
EPI	LDL-C	0.014	0.930	40	1.000
EPI	HDL-C	−0.236	0.143	40	0.936
ST	HbA1c	0.078	0.632	40	1.000
ST	TC	0.162	0.317	40	0.960
ST	TG	0.092	0.572	40	1.000
ST	TyG	0.106	0.516	40	1.000
**ST**	**TyG-BMI**	**0.312**	**0.050**	**40**	**0.935**
ST	TyG-WC	−0.149	0.360	40	0.963
ST	TyG-WHtR	−0.163	0.315	40	0.958
ST	TG/HDL-C	−0.103	0.529	40	1.000
ST	LDL-C	0.128	0.432	40	1.000
ST	HDL-C	0.179	0.268	40	0.936

**Table 10 ijms-26-10068-t010:** Exploratory Spearman correlations between neurotransmitters and metabolic/lipid variables in the T2DM Group (*p* < 0.05, q ≥ 0.05).

Neurotransmitter	Variable	Spearman-ho	*p*-Value	N	q-Value
**DA**	**HbA1c**	**0.269**	**0.024**	**70**	**0.199**
**DA**	**TC**	**0.290**	**0.015**	**70**	**0.176**
DA	TG	0.187	0.121	70	0.689
DA	TyG	0.180	0.136	70	0.758
DA	TyG-BMI	0.135	0.265	70	0.884
DA	TyG-WC	0.035	0.772	70	0.997
DA	TyG-WHtR	0.051	0.675	70	0.996
DA	TG/HDL-C	0.191	0.114	70	0.681
**DA**	**LDL-C**	**0.288**	**0.016**	**70**	**0.195**
DA	HDL-C	−0.060	0.622	70	0.987
NE	HbA1c	0.152	0.209	70	0.772
NE	TC	0.067	0.581	70	0.984
NE	TG	−0.055	0.652	70	0.993
NE	TyG	0.019	0.873	70	1.000
NE	TyG-BMI	−0.008	0.946	70	1.000
NE	TyG-WC	0.031	0.798	70	1.000
NE	TyG-WHtR	0.025	0.837	70	1.000
NE	TG/HDL-C	−0.032	0.795	70	0.998
NE	LDL-C	0.091	0.454	70	0.964
NE	HDL-C	0.035	0.771	70	0.996
EPI	HbA1c	−0.123	0.310	70	0.936
EPI	TC	−0.085	0.482	70	0.970
EPI	TG	−0.014	0.907	70	1.000
EPI	TyG	−0.028	0.816	70	1.000
EPI	TyG-BMI	−0.157	0.193	70	0.772
**EPI**	**TyG-WC**	**−0.365**	**0.002**	**70**	**0.076**
**EPI**	**TyG-WHtR**	**−0.311**	**0.009**	**70**	**0.158**
EPI	TG/HDL-C	0.084	0.491	70	0.976
EPI	LDL-C	−0.086	0.479	70	0.967
EPI	HDL-C	−0.164	0.174	70	0.760
ST	HbA1c	−0.067	0.582	70	0.985
ST	TC	−0.017	0.886	70	1.000
ST	TG	−0.096	0.428	70	0.946
ST	TyG	−0.091	0.454	70	0.964
ST	TyG-BMI	0.004	0.976	70	1.000
ST	TyG-WC	0.078	0.524	70	0.980
ST	TyG-WHtR	0.056	0.646	70	0.989
ST	TG/HDL-C	−0.124	0.308	70	0.884
ST	LDL-C	0.006	0.963	70	1.000
ST	HDL-C	0.095	0.433	70	0.952

**Table 11 ijms-26-10068-t011:** Characteristics of ELISA Kits Used for Neurotransmitter Quantification.

Analyte	Catalog No.	Detection Range	Sensitivity	Intra-/Inter-Assay CV (%)	Reported Cross-Reactivity
Dopamine	E-EL-0046	31.25–2000 pg/mL	18.75 pg/mL	<10	No significant
Norepinephrine	E-EL-0047	0.31–20 ng/mL	0.19 ng/mL	<10	No significant
Epinephrine	E-EL-0045	31.25–2000 pg/mL	18.75 pg/mL	<10	No significant
Serotonin	E-EL-0033	15.63–1000 ng/mL	9.38 ng/mL	<10	No significant

**Table 12 ijms-26-10068-t012:** Calibration curves for each ELISA experiment.

Calibration Curves									
		DA	EPI	ST			NE		
	pg/mL	OD	OD		ng/mL	OD		pg/mL	OD
STD 1	2000	0.169	0.126	STD 1	1000	0.107	STD 1	20	0.096
STD 2	1000	0.338	0.299	STD 2	500	0.280	STD 2	10	0.195
STD 3	500	0.510	0.647	STD 3	250	0.501	STD 3	5	0.333
STD 4	250	0.822	0.932	STD 4	125	0.968	STD 4	2.5	0.756
STD 5	125	1.430	1.984	STD 5	62.5	1.817	STD 5	1.25	1.316
STD 6	62.5	2.062	2.272	STD 6	31.25	2.370	STD 6	0.63	2.557

## Data Availability

The data used to support the findings of this study are available from the corresponding authors upon reasonable request.

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
