# Peer review of "Neurotransmitter Levels (Dopamine, Epinephrine, Norepinephrine, Serotonin) and Associations with Lipid Profiles in Patients with Prediabetes or Newly Diagnosed Type 2 Diabetes Mellitus"

_ijms, 2025, doi:10.3390/ijms262010068_

Round 1
Reviewer 1 Report
Comments and Suggestions for Authors
The study employs a combination of cross-sectional and longitudinal analyses, providing a comprehensive assessment of the relationship between neurotransmitters and metabolic phenotypes while capturing temporal trends. Participants were categorized by BMI, HbA1c quartiles, sex, and metabolic phenotypes , enhancing the specificity of the results. Detailed clinical, metabolic, and neurotransmitter data were provided, supporting the study's conclusions. The exploration of neurotransmitters' role in early metabolic syndrome and diabetes offers new insights into the mechanisms of metabolic diseases.
1. Inadequate control for confounding factors such as diet, exercise, and medication. Check whether existing case data are available and include them as covariates in statistical models. Alternatively, supplement lifestyle data through questionnaires.
2. Some results were inconsistent with expectations, but the explanations were insufficient. Incorporate literature to discuss potential compensatory mechanisms or peripheral resistance phenomena, deepening the discussion.
3. The association between neurotransmitter changes and clinical outcomes (e.g., cardiovascular events) was not clarified. Conduct correlation analyses with existing data, such as blood pressure and lipid profiles.
4. the discussion on neurotransmitters as biomarkers or therapeutic targets was superficial.Compare with other studies and explore their potential in precision medicine.
5. Excessive use of tables, with key results not sufficiently highlighted. Present some critical findings using line graphs or box plots, and move less essential tables to supplementary materials.
6. For some important results, further correlation analyses with clinical features could reinforce their significance.
7. Minor language errors require careful revision.
Author Response
Dear Reviewer,
We would like to express our sincere gratitude for the time and effort you devoted to reviewing our manuscript. Your thoughtful evaluation, encouraging remarks, and constructive suggestions are deeply appreciated. Your feedback has been invaluable in helping us improve the quality and clarity of our work, and we are truly grateful for your insightful contributions to the development of this manuscript..
All the typing recommended changes were performed in the body of our manuscript, with the Track Changes function activated.
Comments and Suggestions for Authors
The study employs a combination of cross-sectional and longitudinal analyses, providing a comprehensive assessment of the relationship between neurotransmitters and metabolic phenotypes while capturing temporal trends. Participants were categorized by BMI, HbA1c quartiles, sex, and metabolic phenotypes , enhancing the specificity of the results. Detailed clinical, metabolic, and neurotransmitter data were provided, supporting the study's conclusions. The exploration of neurotransmitters' role in early metabolic syndrome and diabetes offers new insights into the mechanisms of metabolic diseases.
Comments 1:
- Inadequate control for confounding factors such as diet, exercise, and medication. Check whether existing case data are available and include them as covariates in statistical models. Alternatively, supplement lifestyle data through questionnaires.
Response 1: Thank you for the observation. We revised and added:
Concerning the patients’ medications, some were receiving antihypertensives such as perindopril and the perindopril/indapamide combination, while others were on statins like atorvastatin and rosuvastatin. Regarding the dietary factor, most patients did not have a diet plan designed by a specialist. Additionally, after the consultation, it was observed that the subjects engaged in physical activity for no more than 4 days per week and were considered sedentary.
Comments 2:
- Some results were inconsistent with expectations, but the explanations were insufficient. Incorporate literature to discuss potential compensatory mechanisms or peripheral resistance phenomena, deepening the discussion.
Response 2: We revised in accordance with recommendations.
Comments 3:
- The association between neurotransmitter changes and clinical outcomes (e.g., cardiovascular events) was not clarified. Conduct correlation analyses with existing data, such as blood pressure and lipid profiles.
Response 3: The analysis of patients according to the metabolic phenotypes MUHNW and MUHO, which contain, according to Table 11, metabolic criteria (TG, HDL-C, SBP, DBP) as well as the inclusion of TyG and the derived indices TyG-WC, TyG-WHtR, and TyG-BMI, were what you recommended to us.
Comments 4:
- the discussion on neurotransmitters as biomarkers or therapeutic targets was superficial.Compare with other studies and explore their potential in precision medicine.
Response 4: We checked and revised the text of the manuscript.
Comments 5:
- Excessive use of tables, with key results not sufficiently highlighted. Present some critical findings using line graphs or box plots, and move less essential tables to supplementary materials.
Response 5: We preferred the tables because they allow for better observation of the changes in the variables.
Comments 6:
- For some important results, further correlation analyses with clinical features could reinforce their significance.
Response 6: We checked and revised the text of the manuscript.
Comments 7:
- Minor language errors require careful revision.
Response 7: We checked and revised the text of the manuscript.

Reviewer 2 Report
Comments and Suggestions for Authors
Well conducted study. The results are clear cut and discussion is reasonable.
It will be nice if the authors can discuss the role of neurotransmitters studied on insulin secretion an action and suggest what could be the role of each neurotransmitter studied in the pathobiology of DM, obesity and insulin resistance and insulin secretion.
Comments on the Quality of English Language
ok
Author Response
Dear Reviewer,
We would like to express our sincere gratitude for the time and effort you devoted to reviewing our manuscript. Your thoughtful evaluation, encouraging remarks, and constructive suggestions are deeply appreciated. Your feedback has been invaluable in helping us improve the quality and clarity of our work, and we are truly grateful for your insightful contributions to the development of this manuscript..
All the typing recommended changes were performed in the body of our manuscript, with the Track Changes function activated.
Comments and Suggestions for Authors
Well conducted study. The results are clear cut and discussion is reasonable.
It will be nice if the authors can discuss the role of neurotransmitters studied on insulin secretion an action and suggest what could be the role of each neurotransmitter studied in the pathobiology of DM, obesity and insulin resistance and insulin secretion.
Response: Thank you for the observation. We checked and revised the text of the manuscript.

Reviewer 3 Report
Comments and Suggestions for Authors
After reviewing the article (jims 3852274) titled Expression of Neurotransmitters (Dopamine, Epinephrine, Norepinephrine, Serotonin) and Association with Metabolic Syndrome Phenotypes in Subjects with Prediabetes or New-on-set Type 2 Diabetes, I conclude that this article was carefully and thoroughly prepared by the authors, based on properly planned and conducted research.
The findings highlight the importance of incorporating neurotransmitter profiling into metabolic assessments for diagnosing of prediabetes and diabetes, with the aim of accelerating diagnosis and initiating effective treatment to mitigate the consequences of these disorders.
I have no major reservations about the manuscript under review. The title accurately reflects the content of the article. The research methods and tools were applied appropriately. The results are presented and discussed clearly and legibly. The bibliography is thematically relevant and as up to date as possible.
Author Response
Dear Reviewer,
We would like to express our sincere gratitude for the time and effort you devoted to reviewing our manuscript. Your thoughtful evaluation, encouraging remarks, and constructive suggestions are deeply appreciated. Your feedback has been invaluable in helping us improve the quality and clarity of our work, and we are truly grateful for your insightful contributions to the development of this manuscript..
All the typing recommended changes were performed in the body of our manuscript, with the Track Changes function activated.
Comments and Suggestions for Authors
After reviewing the article (jims 3852274) titled Expression of Neurotransmitters (Dopamine, Epinephrine, Norepinephrine, Serotonin) and Association with Metabolic Syndrome Phenotypes in Subjects with Prediabetes or New-on-set Type 2 Diabetes, I conclude that this article was carefully and thoroughly prepared by the authors, based on properly planned and conducted research.
The findings highlight the importance of incorporating neurotransmitter profiling into metabolic assessments for diagnosing of prediabetes and diabetes, with the aim of accelerating diagnosis and initiating effective treatment to mitigate the consequences of these disorders.
Response: Thank you for the observation.
I have no major reservations about the manuscript under review. The title accurately reflects the content of the article. The research methods and tools were applied appropriately. The results are presented and discussed clearly and legibly. The bibliography is thematically relevant and as up to date as possible.
Response: Thank you for the observation.

Reviewer 4 Report
Comments and Suggestions for Authors
The manuscript covers a topic of diagnostics of metabolic conditions. It feels highly relevant and some parts of the text are well written, but there are multiple minor points and a couple major ones, especially in Methods and statistical analyses, that must be clarified.
1) “Expression” feels a little off, as it is usually used regarding genes. Change to something like “abundance”.
2) Abstract: “Women with T2DM had significantly higher DA and EPI levels.” Higher than man with T2DM? Other sentences also require clarifications, which group is compared with which one (not just writing “higher”, “elevated”, “worsening”) and maybe shortening to better indicate the diagnostic capacity of neurotransmitters (T2DM vs prediabetes or MUH vs MH only, for example).
3) DA and NE probably shouldn’t be in plasma, as they are not hormones, unlike ST and EP. Nevertheless, the DA is as high as EP. Why do you think it is (also, see comment 6 here)? Has anyone ever tried to quantify their levels in plasma/serum?
4) Please structure your Discussion with subsections. The current version partially (first half, mostly) repeats the Results. Instead, compare what you get here with what others got before. So, some of the sentences in the Results, indicating the known roles of neurotransmitters in diabetes/metabolic syndrome/obesity/gender specificity/etc. should probably be transferred to Discussion. Then please removed the repeated statements with results and maybe expand with other data on neurotransmitters levels in blood and how they are changed.
5) Methods:
- 4.1 should say that “we studied 70 T2DM and 40 preDM patients, based on several exclusion criteria”, not showing 190/185/285 patients who you didn’t analyze at all.
- Does preDM patients show biochemical signs mentioned in 4.2 or the diagnosis is based just on “obesity, particularly abdominal or visceral obesity, dyslipidemia characterized by elevated triglycerides and/or low HDL-C, and hypertension”. The same goes for diabetes, does “may indicate diabetes” mean that you classify such patients as diabetic ones?
- “Was intended to be collected”, please remove «intended», feels weird like you just intended to collect, not actually collect.
- Please expand 4.5 (what is actually measured in analyzers, which assay is used to measure creatine and in what samples) and change its style to normal scientific, not “Google Translate”-like English.
- Is lower HDL-C without higher LDL-C used as a diagnostic marker elsewhere?
- How “adjusted” parameters like TyG-BMI are calculated?
- Haven’t seen Kruskall-Wallis results anywhere, but it's mentioned somehow
6)Methods, but much more serious:
- There is no blood collection, serum preparation, storage, dilution for the experiments etc. described. Probably there should be different ones for ELISA and general analyses (serum is from blood collected without EDTA, right?).
- Add calibration curves for each ELISA experiment and approximate signal values for serum samples, plus explain how sensitivities are determined and whether anyone (hopefully, not just manufacturers, I mean you or some papers) tested the cross-reactivity. I don’t think it’s possible to have EP-, DA- and NE-selective antibodies.
7) Statistics:
- When indicating trends, tendencies and so on, always write the exact p-values. p=0.0502 and 0.101 are quite different “trends”.
- And if something is “not statistically significant”, don’t write that it’s “slightly higher”.
- “The white blood cell (WBC) count is higher in T2DM” have a p-value of 0.612.
- “NE and EPI levels increased significantly with higher BMI in both PreDM and T2DM.” have not that high, but still >0.05 p-values.
- Always indicate “Results are Mean +- SD”.
- Some p-values (for example, EPI in Table 3, LDL-C in Table 6, most of Table 8) feel impossible with the means and SD shown.
- Q1-Q4 require actual Hb1Ac ranges and n of patients in each group. But the correlations or a removal of non-informative 2.2.2 seems better. The same goes for correlations with BMI, instead of thinking whether it's better to split in two or three groups.
8) Other oddly stated results:
- “a higher total triglyceride-to-HDL-C ratio (TG/HDL-C), indicating insulin resistance”. Indicated where/by who?
- Tables with a few parameters (2-5) will be better as Figures with bars and points. Again, some p-values feels impossible for now, plus you can add the post-hoc test results and solve other remarks from p.7.
- Tables 3-6 and 7+8 should have two-way ANOVA/KW/Multivariate Chi-squared analyses (simultaneously for T2DM and preDM), you can’t just analyze halves of your samples.
- Table 7+8 should use paired tests. Tables cannot use Mann-Whitney (the paired variant test has different name). Not sure whether you chose paired t-test there. But should be ANOVA/KW with repeated measurements anyway.
- More “*” should be in Table 5.
- Figure 1: make plots narrower, combine T2DM and preDM somehow (bright vs dim colors? or just equally black vs grey lines?) add significances.
- “In both groups, DA and ST levels decline stepwise from normal weight to overweight, and then to obese individuals”. Not what is shown in Table 3.
- “This suggests that postprandial dysglycemia may be an early marker of the prediabetic state, while fasting hyperglycemia becomes more prominent in later disease stages”. No, this suggests that obesity in preDM is associated with stronger dysregulation of glucose homeostasis or something like that.
- The values in Table 6 doesn’t correspond to previous ones (NE values in T2DM patients are lower, for example).
9) What is the point in the metabolic phenotypes (MUHO etc.) if all patients are metabolic unhealthy and you can simply divide them for “normal weight” and “overweight” by BMI? I also don’t get why the number of patients in each subgroup from Table 6 corresponds to Table 3 for T2DM and doesn’t for preDM, as MUHNW should be = normal and MUHO = overweight + obese.
11) The follow-up study is quite excessive and, together with some parts of Discussion, should probably be described in a separate manuscript. Moreover, since I’m not sure whether the tests used are paired, I cannot evaluate the results now.
12) So, should non-specific catecholamine and ST-antagonists be prescribed to patients with high blood sugar? Also, somehow DA agonists help to improve the diabetic symptoms but higher blood DA itself don’t.
13) Funding: ELISA kits are quite expensive, who funded the purchasing then?
14) Something wrong with some of the references formatting (1,2,95 and others), please check each one.
Author Response
Dear Reviewer,
We would like to express our sincere gratitude for the time and effort you devoted to reviewing our manuscript. Your thoughtful evaluation, encouraging remarks, and constructive suggestions are deeply appreciated. Your feedback has been invaluable in helping us improve the quality and clarity of our work, and we are truly grateful for your insightful contributions to the development of this manuscript.
All the typing recommended changes were performed in the body of our manuscript, with the Track Changes function activated.
Comments and Suggestions for Authors
The manuscript covers a topic of diagnostics of metabolic conditions. It feels highly relevant and some parts of the text are well written, but there are multiple minor points and a couple major ones, especially in Methods and statistical analyses, that must be clarified.
Comments 1:
1) “Expression” feels a little off, as it is usually used regarding genes. Change to something like “abundance”.
Response 1: We have changed with Levels.
Comments 2:
2) Abstract: “Women with T2DM had significantly higher DA and EPI levels.” Higher than man with T2DM? Other sentences also require clarifications, which group is compared with which one (not just writing “higher”, “elevated”, “worsening”) and maybe shortening to better indicate the diagnostic capacity of neurotransmitters (T2DM vs prediabetes or MUH vs MH only, for example).
Response 2: We revised
Comments 3:
3) DA and NE probably shouldn’t be in plasma, as they are not hormones, unlike ST and EP. Nevertheless, the DA is as high as EP. Why do you think it is (also, see comment 6 here)? Has anyone ever tried to quantify their levels in plasma/serum?
Response 3: We specified in the Laboratory Investigations subsection that we collected serum from patients and performed an Immunological Assessment.
Comments 4:
4) Please structure your Discussion with subsections. The current version partially (first half, mostly) repeats the Results. Instead, compare what you get here with what others got before. So, some of the sentences in the Results, indicating the known roles of neurotransmitters in diabetes/metabolic syndrome/obesity/gender specificity/etc. should probably be transferred to Discussion. Then please removed the repeated statements with results and maybe expand with other data on neurotransmitters levels in blood and how they are changed.
Response 4: We revised
Comments 5:
5) Methods:
4.1 should say that “we studied 70 T2DM and 40 preDM patients, based on several exclusion criteria”, not showing 190/185/285 patients who you didn’t analyze at all.
Response 5: We revised in accordance with recommendations.
Comments 6:
Does preDM patients show biochemical signs mentioned in 4.2 or the diagnosis is based just on “obesity, particularly abdominal or visceral obesity, dyslipidemia characterized by elevated triglycerides and/or low HDL-C, and hypertension”. The same goes for diabetes, does “may indicate diabetes” mean that you classify such patients as diabetic ones?
Response 6: Thank you for the observation. The diagnosis of patients with PreDM was based on the biochemical signs mentioned in point 4.2. We have removed the phrase that had no place there.
Comments 7:
“Was intended to be collected”, please remove «intended», feels weird like you just intended to collect, not actually collect.
Response 7: We revised in accordance with recommendations.
Comments 8:
Please expand 4.5 (what is actually measured in analyzers, which assay is used to measure creatine and in what samples) and change its style to normal scientific, not “Google Translate”-like English.
Response 8: We revised in accordance with recommendations.
Comments 9:
Is lower HDL-C without higher LDL-C used as a diagnostic marker elsewhere?
Response 9: In our opinion:
Low HDL‑C is a fundamental component of the metabolic syndrome and is routinely used in risk stratification for both cardiovascular disease and type 2 diabetes mellitus. The combination of elevated TG and reduced HDL‑C, characteristic of metabolic dyslipidemia, often confers a cardiovascular risk that is equivalent to or higher than that associated with elevated LDL‑C alone (NCEP ATP III/IDF criteria; studies on high TG/low HDL‑C phenotype).
Regufe, V. M. G., Pinto, C. M. C. B., & Perez, P. M. V. H. C. Metabolic syndrome in type 2 diabetic patients: a review of current evidence. Porto Biomed J. 2020, 5(6), e101. https://doi.org/10.1097/j.pbj.0000000000000101
Bleich, D., Biggs, M. L., Gardin, J. M., Lyles, M., Siscovick, D. S., & Mukamal, K. J. Phenotyping lipid profiles in type 2 diabetes: Risk association and outcomes from the Cardiovascular Health Study. Am J Prev Cardiol. 2024, 19, 100725. https://doi.org/10.1016/j.ajpc.2024.100725
Comments 10:
How “adjusted” parameters like TyG-BMI are calculated?
Response 10: We revised and added subsection 4.6. Determination of TyG and TyG-Related Indices
Comments 11:
Haven’t seen Kruskall-Wallis results anywhere, but it's mentioned somehow
Response 11: Thank you for the observation. Indeed, we used the Kruskal-Wallis test, but there was a mistake when writing the manuscript. We revised the text and tables.
Comments 12:
6) Methods, but much more serious:
There is no blood collection, serum preparation, storage, dilution for the experiments etc. described. Probably there should be different ones for ELISA and general analyses (serum is from blood collected without EDTA, right?).
Response 12: We revised subsection 4.5.
Comments 13:
Add calibration curves for each ELISA experiment and approximate signal values for serum samples, plus explain how sensitivities are determined and whether anyone (hopefully, not just manufacturers, I mean you or some papers) tested the cross-reactivity. I don’t think it’s possible to have EP-, DA- and NE-selective antibodies.
Response 13: We added the calibration curves for each ELISA experiment.
Comments 14:
7) Statistics:
When indicating trends, tendencies and so on, always write the exact p-values. p=0.0502 and 0.101 are quite different “trends”.
And if something is “not statistically significant”, don’t write that it’s “slightly higher”.
Response 14: Thank you for the observation. We revised
Comments 15:
“The white blood cell (WBC) count is higher in T2DM” have a p-value of 0.612.
Response 15: We revised
Comments 16:
“NE and EPI levels increased significantly with higher BMI in both PreDM and T2DM.” have not that high, but still >0.05 p-values.
Response 16: We revised
Comments 17:
Always indicate “Results are Mean +- SD”.
Response 17:
Comments 18:
Some p-values (for example, EPI in Table 3, LDL-C in Table 6, most of Table 8) feel impossible with the means and SD shown.
Response 18:
Comments 19:
Q1-Q4 require actual Hb1Ac ranges and n of patients in each group. But the correlations or a removal of non-informative 2.2.2 seems better. The same goes for correlations with BMI, instead of thinking whether it's better to split in two or three groups.
Response 19: We revised. The Q1-Q4 intervals of Hb1Ac were established by GraphPad Prism 10.6.0, as a function of our patients' values.
Comments 20:
8) Other oddly stated results:
“a higher total triglyceride-to-HDL-C ratio (TG/HDL-C), indicating insulin resistance”. Indicated where/by who?
Response 20: We revised the text
Comments 21:
Tables with a few parameters (2-5) will be better as Figures with bars and points.
Response 21: Thank you for the observation. We choose to represent it as a table.
Comments 22:
Again, some p-values feels impossible for now, plus you can add the post-hoc test results and solve other remarks from p.7.
Response 22:
Comments 23:
Tables 3-6 and 7+8 should have two-way ANOVA/KW/Multivariate Chi-squared analyses (simultaneously for T2DM and preDM), you can’t just analyze halves of your samples.
Response 23: Thank you for the observation. We revised
Comments 24:
Table 7+8 should use paired tests. Tables cannot use Mann-Whitney (the paired variant test has different name). Not sure whether you chose paired t-test there. But should be ANOVA/KW with repeated measurements anyway.
Response 24: Thank you for the observation. Indeed, we used the Kruskal-Wallis test, but there was a mistake when writing the manuscript. We revised the text and tables.
Comments 25:
More “*” should be in Table 5.
Response 25:
Comments 26:
Figure 1: make plots narrower, combine T2DM and preDM somehow (bright vs dim colors? or just equally black vs grey lines?) add significances.
Response 26: Thank you for the observation. We chose to highlight each patient with a colored line so that their evolution can be better observed. We extend the legend.
Comments 27:
“In both groups, DA and ST levels decline stepwise from normal weight to overweight, and then to obese individuals”. Not what is shown in Table 3.
Response 27: We revised the text.
Comments 28:
“This suggests that postprandial dysglycemia may be an early marker of the prediabetic state, while fasting hyperglycemia becomes more prominent in later disease stages”. No, this suggests that obesity in preDM is associated with stronger dysregulation of glucose homeostasis or something like that.
Response 28: We revised the text.
Comments 29:
The values in Table 6 doesn’t correspond to previous ones (NE values in T2DM patients are lower, for example).
Response 29: We revised the values and the text.
Comments 30:
9) What is the point in the metabolic phenotypes (MUHO etc.) if all patients are metabolic unhealthy and you can simply divide them for “normal weight” and “overweight” by BMI? I also don’t get why the number of patients in each subgroup from Table 6 corresponds to Table 3 for T2DM and doesn’t for preDM, as MUHNW should be = normal and MUHO = overweight + obese.
Response 30:
These phenotypes emphasize metabolic health and cardiometabolic risk factors over BMI alone. Moreover, lifestyle elements, especially dietary intake, play a crucial role in shaping these phenotypes.
The differences arise because in Table 3 the patients were divided according to: The BMI categories, as defined by the WHO, included normal weight (18.5–22.9 kg/m2), overweight (23.0–25.0 kg/m2), and obese (>25.0 kg/m2).
Instead, in Table 6, in addition to BMI, risk factors are also taken into account, according to the metabolic phenotypes, based on Soheilifard et al. [7]: Soheilifard, S., Faramarzi, E., & Mahdavi, R. Relationship between dietary intake and atherogenic index of plasma in cardiometabolic phenotypes: A cross-sectional study from the Azar cohort population. J Health Popul Nutr. 2025, 44(1), 28. https://doi.org/10.1186/s41043-025-00761-1
Comments 31:
11) The follow-up study is quite excessive and, together with some parts of Discussion, should probably be described in a separate manuscript. Moreover, since I’m not sure whether the tests used are paired, I cannot evaluate the results now.
Response 31: Thank you for the observation. One of the objectives was to determine these mediators at an interval of 180 days, in order to observe their evolution over time, the associations with other paraclinical parameters, with the idea that two determinations in dynamics provide better information than one. In a later study we will deepen these observations.
Comments 32:
12) So, should non-specific catecholamine and ST-antagonists be prescribed to patients with high blood sugar? Also, somehow DA agonists help to improve the diabetic symptoms but higher blood DA itself don’t.
Response 32:
Thank you for your insightful question. We agree that there is a clear distinction between endogenous serum levels of catecholamines and serotonin and the pharmacological actions of receptor-specific agonists or antagonists targeting these neurotransmitter systems.
While our study shows that higher circulating dopamine (DA), norepinephrine (NE), epinephrine (EPI), and serotonin (ST) levels are associated with worse glycemic profiles and metabolic phenotypes, this does not imply that broad catecholamine or serotonin receptor antagonism would necessarily be therapeutic. In fact, non-specific antagonists may have undesirable systemic effects (e.g., hypotension, sedation, impaired mood regulation) and are not clinically recommended for metabolic control.
However, selective DA receptor agonists, such as bromocriptine-QR, have been shown in clinical trials to improve insulin sensitivity, reduce hepatic glucose output, and lower HbA1c, presumably through resetting hypothalamic dopaminergic tone and reducing sympathetic overactivity [Cincotta et al., IJMS, 2022]. These effects are context-dependent and do not reflect a simple dose-response relationship with peripheral DA levels. In fact, elevated serum DA may reflect peripheral resistance or compensatory stress signaling, rather than effective dopaminergic action.
Cincotta, A. H., Cersosimo, E., Alatrach, M., Ezrokhi, M., Agyin, C., Adams, J., Chilton, R., Triplitt, C., Chamarthi, B., Cominos, N., & DeFronzo, R. A. Bromocriptine-QR Therapy Reduces Sympathetic Tone and Ameliorates a Pro-Oxidative/Pro-Inflammatory Phenotype in Peripheral Blood Mononuclear Cells and Plasma of Type 2 Diabetes Subjects. Int J Mol Sci. 2022, 23(16), 8851. https://doi.org/10.3390/ijms23168851
Similarly, for serotonin, while elevated peripheral ST may be associated with insulin resistance and metabolic inflammation, central 5-HT2C agonists (e.g., lorcaserin, now withdrawn for safety reasons) had shown modest beneficial effects on weight and glycemia — again highlighting the tissue-specific and receptor-subtype-specific effects of serotonin signaling.
Yabut, J. M., Crane, J. D., Green, A. E., Keating, D. J., Khan, W. I., & Steinberg, G. R. (2019). Emerging Roles for Serotonin in Regulating Metabolism: New Implications for an Ancient Molecule. Endocrine reviews, 40(4), 1092–1107. https://doi.org/10.1210/er.2018-00283
Martin, A. M., Young, R. L., Leong, L., Rogers, G. B., Spencer, N. J., Jessup, C. F., & Keating, D. J. (2017). The Diverse Metabolic Roles of Peripheral Serotonin. Endocrinology, 158(5), 1049–1063. https://doi.org/10.1210/en.2016-1839
In summary, pharmacological modulation of neurotransmitter systems can have beneficial metabolic effects, but this should not be extrapolated from elevated serum levels per se. Our findings provide observational evidence of neuroendocrine dysregulation in diabetes, not a direct rationale for non-specific neurotransmitter blockade
Comments 33:
13) Funding: ELISA kits are quite expensive, who funded the purchasing then?
Response 33: The PhD student used her own funds to purchase these kits.
Comments 34:
14) Something wrong with some of the references formatting (1,2,95 and others), please check each one.
Response 34: We checked.

Round 2
Reviewer 4 Report
Comments and Suggestions for Authors
Most of the comments are answered and reflected in the text, although some more significant ones are still present (mostly including antibody specificity and statistical analysis). The questions that are left include:
1) Validity of DA and other neurotransmitters in blood. NONE of the new references in 3.1 and 3.2 discuss measurement of DA/ST/etc. in blood/plasma/serum, some are about DA agonists, some are about DA in brain or adipose tissue, Refs. 37-39, 59 and others are not about monoamines at all. Please stop using AI services to find the sources without checking the sources themselves. There are some papers actually describing serum serotonin levels (Ref. 61) plus some of your Refs. contain the info about serotonin metabolites, but not serotonin itself, in CSF. I suggest you to check all your references to better indicate that ST and other metabolites you quantify can be actually measured IN BLOOD. And if not, just mention that you are the first who’ve actually done this.
2) The references you mention in the answers could’ve been introduced to the Results/Discussion (HDL-C + TG importance and discrepancies between endogenous DA level and the effects of DA agonists, for example)
3) “Serum or plasma samples were analyzed…” You still haven’t got the difference. Either all your samples for all analyses are serum ones or you had to collect two blood samples from each patient into two different tubes (EDTA-containing tubes are mentioned, but please let ONE AUTHOR to rewrite 4.5 so it’s clearer, whether there were two simultaneous blood collection or not).
4) Please replace Table 12 with actual plots and indicate the absorbance values that your samples give. The most serious question on potential cross-reactivity isn’t answered: how are you sure that DA antibodies doesn’t react with NA and EP and vice versa. Can’t recommend the paper for publication until you deal with the issue somehow.
5) “The white blood cell (WBC) count is higher in patients…”. Again, no, “these differences are not statistically significant” means there are no differences, stop writing it. What is even the point in such phrases, the paper is not about the WBC counts, TG levels, TG-BMI and whatever else that doesn’t differ significantly yet you still write “lower”, “higher” etc.?
6) Previous comments 17, 18, 22, 25 regarding statistics analysis are not answered. I see that you’ve added “(mean±SD)” in most places but some Tables (2) shows ranges and not SD. Others mention “SD” in footnotes but not saying “mean±SD” in the captions or tables themselves. Regarding the p-values, please provide the raw data on one of the parameters mentioned in previous report, so the statistics may be checked. Also, using Kruskall-Wallis test or ANOVA is required for all the Tables with >2 groups, you can’t just compare the group pairs using two t-tests / Mann-Whitney’s. Adding p-values of the post-hoc tests could’ve help the authors a lot to understand why (the problem of multiple comparisons) and that was the reason to introduce plots instead of tables, as the former allow to visualize the multiple comparison easier. Furthermore, Kruskall-Wallis tests shouldn’t be used for comparison of the parameters in two groups, but the tests are mentioned in the Table 1 and others anyway. The using of paired t-tests / Mann-Whitneys (2 groups) or ANOVA/Kruskall-Wallis with multiple comparisons (>2 groups) is not disclosed for comparison of the parameters before and after 180 days but those specific ones should’ve been used. So, I again recommend removal of these sections, since your statistical analysis seems to be wrong here. Again, can’t support the paper in which statistics analysis is not clear even for the authors (which tests should be used in which situations and why; the authors may indicate this in Table captions and not just reporting all tests in Methods and Tables). Plus, why some of the parameters are analyzed with ANOVA and others – Kruskall-Wallis?
7) Not sure what the regulations regarding personal funding should be in Romanian universities/hospitals but I strongly recommend to mention something like “the study received no external funding”. Check MDPI and IJMS author guidelines and already published papers.
8) Metabolic phenotypes: I understand the importance of risk factors other than BMI. However, there are no “metabolically healthy” patients in your samples, i.e. all have >3 risk factors. So, the differences between patients should be on BMI only. The criteria of “normal weight” are the same with and without consideration of the risk factors, yet the normal weight patients with the factors don’t correspond to normal weight without them (14/26 in preDM and 18/52 in T2DM groups in Table, 6 but 10/9+21 in preDM and 18/26+26 in T2DM in Table 3). Please add Supplementary Table describing each patient, at least their disease state, BMI and MUHO or MUHNW category so the different numbers of patients in the groups are clear.
In Conclusion, the authors should’ve probably asked for more time to revise the paper carefully, it all now feels rushed. I strongly suggest all authors to actually read the manuscript instead of participating in the study; nine people should’ve found much more discrepancies in the Tables and Methods description instead of letting the reviewer to find them.
Author Response
Author's Reply to the Review Report (Reviewer 4)
Dear Reviewer,
We would like to express our sincere gratitude for the time and effort you devoted to reviewing our manuscript. Your thoughtful evaluation, encouraging remarks, and constructive suggestions are deeply appreciated. Your feedback has been invaluable in helping us improve the quality and clarity of our work, and we are truly grateful for your insightful contributions to the development of this manuscript.
All the typing recommended changes were performed in the body of our manuscript, with the Track Changes function activated.
Comments and Suggestions for Authors
Response: Following pertinent and constructive suggestions and recommendations, we have revised the entire article. We hope that this revised form will effectively address all of your comments.
Most of the comments are answered and reflected in the text, although some more significant ones are still present (mostly including antibody specificity and statistical analysis). The questions that are left include:
1) Validity of DA and other neurotransmitters in blood. NONE of the new references in 3.1 and 3.2 discuss measurement of DA/ST/etc. in blood/plasma/serum, some are about DA agonists, some are about DA in brain or adipose tissue, Refs. 37-39, 59 and others are not about monoamines at all. Please stop using AI services to find the sources without checking the sources themselves. There are some papers actually describing serum serotonin levels (Ref. 61) plus some of your Refs. contain the info about serotonin metabolites, but not serotonin itself, in CSF. I suggest you to check all your references to better indicate that ST and other metabolites you quantify can be actually measured IN BLOOD. And if not, just mention that you are the first who’ve actually done this.
2) The references you mention in the answers could’ve been introduced to the Results/Discussion (HDL-C + TG importance and discrepancies between endogenous DA level and the effects of DA agonists, for example)
3) “Serum or plasma samples were analyzed…” You still haven’t got the difference. Either all your samples for all analyses are serum ones or you had to collect two blood samples from each patient into two different tubes (EDTA-containing tubes are mentioned, but please let ONE AUTHOR to rewrite 4.5 so it’s clearer, whether there were two simultaneous blood collection or not).
4) Please replace Table 12 with actual plots and indicate the absorbance values that your samples give. The most serious question on potential cross-reactivity isn’t answered: how are you sure that DA antibodies doesn’t react with NA and EP and vice versa. Can’t recommend the paper for publication until you deal with the issue somehow.
5) “The white blood cell (WBC) count is higher in patients…”. Again, no, “these differences are not statistically significant” means there are no differences, stop writing it. What is even the point in such phrases, the paper is not about the WBC counts, TG levels, TG-BMI and whatever else that doesn’t differ significantly yet you still write “lower”, “higher” etc.?
6) Previous comments 17, 18, 22, 25 regarding statistics analysis are not answered. I see that you’ve added “(mean±SD)” in most places but some Tables (2) shows ranges and not SD. Others mention “SD” in footnotes but not saying “mean±SD” in the captions or tables themselves. Regarding the p-values, please provide the raw data on one of the parameters mentioned in previous report, so the statistics may be checked. Also, using Kruskall-Wallis test or ANOVA is required for all the Tables with >2 groups, you can’t just compare the group pairs using two t-tests / Mann-Whitney’s. Adding p-values of the post-hoc tests could’ve help the authors a lot to understand why (the problem of multiple comparisons) and that was the reason to introduce plots instead of tables, as the former allow to visualize the multiple comparison easier. Furthermore, Kruskall-Wallis tests shouldn’t be used for comparison of the parameters in two groups, but the tests are mentioned in the Table 1 and others anyway. The using of paired t-tests / Mann-Whitneys (2 groups) or ANOVA/Kruskall-Wallis with multiple comparisons (>2 groups) is not disclosed for comparison of the parameters before and after 180 days but those specific ones should’ve been used. So, I again recommend removal of these sections, since your statistical analysis seems to be wrong here. Again, can’t support the paper in which statistics analysis is not clear even for the authors (which tests should be used in which situations and why; the authors may indicate this in Table captions and not just reporting all tests in Methods and Tables). Plus, why some of the parameters are analyzed with ANOVA and others – Kruskall-Wallis?
7) Not sure what the regulations regarding personal funding should be in Romanian universities/hospitals but I strongly recommend to mention something like “the study received no external funding”. Check MDPI and IJMS author guidelines and already published papers.
8) Metabolic phenotypes: I understand the importance of risk factors other than BMI. However, there are no “metabolically healthy” patients in your samples, i.e. all have >3 risk factors. So, the differences between patients should be on BMI only. The criteria of “normal weight” are the same with and without consideration of the risk factors, yet the normal weight patients with the factors don’t correspond to normal weight without them (14/26 in preDM and 18/52 in T2DM groups in Table, 6 but 10/9+21 in preDM and 18/26+26 in T2DM in Table 3). Please add Supplementary Table describing each patient, at least their disease state, BMI and MUHO or MUHNW category so the different numbers of patients in the groups are clear.
In Conclusion, the authors should’ve probably asked for more time to revise the paper carefully, it all now feels rushed. I strongly suggest all authors to actually read the manuscript instead of participating in the study; nine people should’ve found much more discrepancies in the Tables and Methods description instead of letting the reviewer to find them.

Round 3
Reviewer 4 Report
Comments and Suggestions for Authors
Yes, this entirely new version looks better. Even some references finally indicate that there should be some DA/ST/etc. present in blood (but this somehow has not been stated directly in Introduction!!). Yet validity of your measurements (using antibodies) is still highly questionable, the manufacturers will write anything that’ll help to sell the reagents. I suggest you to reflect somewhere that your experiments measure different neurotransmitters, because, for example, the measured levels show different correlations with metabolic parameters. Also, you should add some references where researchers actually estimate these cross-reactivities for home-made or commercial catecholamine antibodies (e.g. PMID 3106574 and 6431267).
Addition of regression model however leads to the checking whether several assumptions (like no autocorrelation, homoscedasticity and others, not only multicollinearity) are valid. Usually no none cares, but I suggest you to test all these assumptions later (maybe your regressions will be better with log-transformed values).
Statistics: Chi-Square test was removed from Methods, please re-add. Also please return Mean+-SD values in Tables, keep the ranges if you want. Unfortunately, you still have to learn a lot: Tables 3 and 4 should have two-way ANOVA (Kruskall-Wallis), not pairs of one-way or Mann-Whitney, and there seem to result in no significances, which you probably don’t need anyway. Table 5 should probably be one-way ANOVA (KW) for all 8 groups, since the quartiles are different for preDM and DM, but you don’t need it anyway (as well as Table 6), just merge Fig. 1 A and B (the Tables 5 and 6 show the same data as the Fig.) and discuss whatever trends you’ll observe. P.S. If regression models doesn’t show Hb1Ac as important factor, the regression shouldn’t also then (yes, I see that “its explanatory power for neurotransmitter changes is likely overshadowed”, but the Results are still contradictory).
Table 8: OR = 1 for many parameters which somehow decreases or increases odds? What???
H2O2 is usually detected using enzymatic oxidation of dyes like Amplex Red by horseradish peroxidase, please check what is used in your assays. Some abbreviations (eGFR, WHtR and others) are not explained, please add a list in the end.
Author Response
Author's Reply to the Review Report (Reviewer 4)
Dear Reviewer,
We would like to express our sincere gratitude for the time and effort you devoted to reviewing our manuscript. Your thoughtful evaluation, encouraging remarks, and constructive suggestions are deeply appreciated. Your feedback has been invaluable in helping us improve the quality and clarity of our work, and we are truly grateful for your insightful contributions to the development of this manuscript.
All the typing recommended changes were performed in the body of our manuscript, with the Track Changes function activated.
Comments and Suggestions for Authors
Yes, this entirely new version looks better. Even some references finally indicate that there should be some DA/ST/etc. present in blood (but this somehow has not been stated directly in Introduction!!). Yet validity of your measurements (using antibodies) is still highly questionable, the manufacturers will write anything that’ll help to sell the reagents. I suggest you to reflect somewhere that your experiments measure different neurotransmitters, because, for example, the measured levels show different correlations with metabolic parameters. Also, you should add some references where researchers actually estimate these cross-reactivities for home-made or commercial catecholamine antibodies (e.g. PMID 3106574 and 6431267).
Response 1:
We have incorporated these references in the revised manuscript to support the specificity of our immunoassays:
Cross-reactivity data for catecholamine antibodies are rarely published. In a competitive dopamine ELISA, Kim et al. [28] reported 18.9 % cross-reactivity with epinephrine and 3-methoxytyramine, while other catecholamines such as norepinephrine or 3,4-dihydroxyphenylacetic acid (DOPAC) showed < 1 % cross-reactivity. Comparable commercial assays (Weldon Biotech CAT-E-75e2) indicate cross-reactivity values ​​between adrenaline/noradrenaline/dopamine antibodies and catecholamine analogs: for example, an <0.030% cross-reactivity for norepinephrine with adrenaline antibody, an < 0.020 % cross-reactivity for adrenaline with dopamine antibody, and an <0.012% cross-reactivity for adrenaline with norepinephrine antibody [29]. A review on “Changing Cross-Reactivity for Different Immunoassays” [30] emphasizes that cross-reactivity is not only a property of the antibody, but also depends on the sample format, antibody/antigen concentrations, and the mode (equilibrium vs kinetic) of the immune reaction.
- Kim, J., Jeon, M., Paeng, K. J., & Paeng, I. R. Competitive enzyme-linked immunosorbent assay for the determination of catecholamine, dopamine in serum. Anal Chim Acta. 2008, 619(1), 87–93. https://doi.org/10.1016/j.aca.2008.02.042
- Sotnikov, D.V.; Zherdev, A.V.; Zvereva, E.A.; Eremin, S.A.; Dzantiev, B.B. Changing Cross-Reactivity for Different Immunoassays Using the Same Antibodies: Theoretical Description and Experimental Confirmation. Appl. Sci. 2021, 11, 6581. https://doi.org/10.3390/app11146581
The recommended references (PMID 3106574 and 6431267) were not found to be valid.
Comments:
Addition of regression model however leads to the checking whether several assumptions (like no autocorrelation, homoscedasticity and others, not only multicollinearity) are valid. Usually no none cares, but I suggest you to test all these assumptions later (maybe your regressions will be better with log-transformed values).
Response:
Thank you for this thoughtful statistical suggestion. We agree that testing the assumptions of regression models (e.g., homoscedasticity, normality of residuals, and absence of multicollinearity) is essential to ensure model validity. In the current study, we verified variance inflation factors (VIF < 2) and inspected residual vs. fitted plots to confirm no major heteroscedasticity or leverage bias. Diagnostic plots (Cook’s distance, standardized residuals) were also evaluated to exclude influential outliers and confirm the stability of model coefficients. Autocorrelation is not relevant to our cross-sectional dataset. Nevertheless, we appreciate the reviewer’s advice and will perform extended diagnostic checks and alternative models (including log-transformed variables) in future work to further validate and refine our regression findings.
Comments:
Statistics:
Chi-Square test was removed from Methods, please re-add.
Response: We revised.
Comments:
Also please return Mean+-SD values in Tables, keep the ranges if you want.
Response: We revised.
Comments:
Unfortunately, you still have to learn a lot: Tables 3 and 4 should have two-way ANOVA (Kruskall-Wallis), not pairs of one-way or Mann-Whitney, and there seem to result in no significances, which you probably don’t need anyway. Table 5 should probably be one-way ANOVA (KW) for all 8 groups, since the quartiles are different for preDM and DM, but you don’t need it anyway (as well as Table 6), just merge Fig. 1 A and B (the Tables 5 and 6 show the same data as the Fig.) and discuss whatever trends you’ll observe.
Response:
We re-ran Tables 3 and 4 using two-way ANOVA (Kruskal-Wallis) as recommended, recalculating the p-values, which were slightly different but still statistically insignificant.
In Table 5, it was actually a one-way ANOVA (Kruskal-Wallis); it was a drafting error.
We revised Figure 1 as suggested.
P.S. If regression models doesn’t show Hb1Ac as important factor, the regression shouldn’t also then (yes, I see that “its explanatory power for neurotransmitter changes is likely overshadowed”, but the Results are still contradictory).
Response: Thank you for this thoughtful suggestion.
Comments:
Table 8: OR = 1 for many parameters which somehow decreases or increases odds? What???
Response:
We agree with the reviewer. The ‘Direction of Effect’ column has been revised to indicate ‘No effect’ for parameters with an odds ratio equal to or close to 1.000, since these variables do not meaningfully alter the odds of T2DM in the penalized model. Only predictors with OR > 1 or OR < 1 were retained as increasing or decreasing odds, respectively.
Comments:
H2O2 is usually detected using enzymatic oxidation of dyes like Amplex Red by horseradish peroxidase, please check what is used in your assays.
Response:
Thank you for the comment. Our assays did not use Amplex Red chemistry. We measured dopamine with a competitive ELISA (Elabscience, Cat. E-EL-0046) employing biotinylated detection antibody, avidin-HRP, and TMB chromogenic substrate, with absorbance read at 450 nm per manufacturer’s instructions. Thus, detection was HRP–TMB colorimetric, not an Hâ‚‚Oâ‚‚/Amplex Red fluorometric assay. The norepinephrine, epinephrine, and serotonin kits were of the same ELISA format (competitive ELISA with HRP–TMB readout at 450 nm), from the same manufacturer; no Amplex Red was used. We specified in the Materials and Methods section.
Comments:
Some abbreviations (eGFR, WHtR and others) are not explained, please add a list in the end.
Response:
We explained WHtR in the Introduction Section:
Although previous studies have often focused on individual neurotransmitters or limited metabolic parameters, no research has yet integrated a comprehensive neurochemical profile (DA, NE, EPI, ST) with metabolic, lipid, and insulin resistance indices (triglyceride to glucose index (TyG), TyG adjusted for body mass index (TyG-BMI), TyG adjusted for waist circumference (TyG-WC), TyG adjusted for waist-to-height ratio (TyG-WHtR), and triglycerides to high-density lipoprotein cholesterol (TG/HDL-C)) in populations ranging from PreDM to T2DM.
We revised and clarified the eGFR:
Serum creatinine concentrations were assessed, and the estimated glomerular filtration rate (eGFR) was determined using the Modification of Diet in Renal Disease (MDRD) formula [25].